# Remote detection of radioactive material using high-power pulsed electromagnetic radiation

Dongsung Kim[1], Dongho Yu[1], Ashwini Sawant[2], Mun Seok Choe[1], Ingeun Lee[2], Sung Gug Kim[2] & EunMi Choi[1]

Remote detection of radioactive materials is impossible when the measurement location is far from the radioactive source such that the leakage of high-energy photons or electrons from the source cannot be measured. Current technologies are less effective in this respect because they only allow the detection at distances to which the high-energy photons or electrons can reach the detector. Here we demonstrate an experimental method for remote detection of radioactive materials by inducing plasma breakdown with the high-power pulsed electromagnetic waves. Measurements of the plasma formation time and its dispersion lead to enhanced detection sensitivity compared to the theoretically predicted one based only on the plasma on and off phenomena. We show that lower power of the incident electromagnetic wave is sufficient for plasma breakdown in atmospheric-pressure air and the elimination of the statistical distribution is possible in the presence of radioactive material.

[1] Department of Physics, School of Natural Science, Ulsan National Institute of Science and Technology, Ulsan 44919, South Korea. [2] Department of Electrical and Computer Engineering, Ulsan National Institute of Science and Technology, Ulsan 44919, South Korea. Correspondence and requests for materials should be addressed to E.M.C. (email: emchoi@unist.ac.kr).

Threats owing to man-made radioactivity, including accidents at nuclear power plants and nuclear weapons, have increased and are unavoidable. Once such an accident has occurred, the accident site should be closed to human beings. Therefore, the ability to detect radioactive material remotely is essential to protect not only the residents in the areas near accident sites, but also those working to dismantle the malfunctioning systems. Further, in the case of radioactive material being smuggled via marine transportation, it is difficult to detect the presence of radioactive cargo before it has been offloaded at the port, since the current technology is incapable of the remote detection of radioactive material. In addition, radiological dispersal devices (so-called dirty bombs) have become a serious threat to national security. For instance, the public was recently shocked by reports of a radioactive drone (http://www.bbc.com/news/world-asia-32465624). To check for the presence of radioactive material in a drone, the ability to detect radioactive material remotely is essential. While the conventional methods and devices used to detect radioactive material, such as Geiger–Muller counters and ion chamber detectors, are convenient, they have technical limits when it comes to detecting unconfirmed radioactive sources at large distances. A typical Geiger–Muller counter may be capable of measuring 1 milliCurie (mCi) of Cobalt-60 ($^{60}$Co) (with a gamma ray energy $> 7\,keV$) with a resolution of $\sim 1\,\mu Sv\,h^{-1}$ at a maximum distance of $\sim 3.5\,m$ away from the source (Supplementary Note 1). However, when it is necessary to measure lower levels of radioactivity or to sense the material at distances much longer than tens of metres, these conventional devices have limitations. Consequently, scientists have attempted to overcome these technical limitations. For instance, a laser-induced plasma was proposed for use in the remote detection of radioactive material based on the reduction of the delay time before plasma formation[1,2]. However, when using this method, multiphoton ionization can occur in the absence of the free electrons that are generated by radioactive materials, which raises significant reliability-related issues. A method proposed recently by researchers at the University of Maryland involves the use of a high-intensity millimetre- or terahertz (THz)-wave source to sense radioactivity[3]. A focused high-power millimetre/THz-wave source with an electric field well above the threshold causes plasma breakdown when at least one seed electron is present in the volume of the focused beam to initiate the avalanche breakdown process[4–7]. If the millimetre/THz wave is focused near a radioactive source, plasma breakdown occurs immediately due to the exponential increase in the number of free electrons[3,8]. It is known that, in ambient air, a few (or tens of) free electrons exist per cubic centimetre. When a millimetre/THz wave is focused on a very small spot in which almost no electrons are likely to be present, the plasma discharge signal coming from the area on which the beam is focused can be considered to originate from a nearby radioactive source. However, the method of remotely detecting radioactive materials using a high-power millimetre/THz-wave source has not yet been demonstrated or investigated experimentally with a real radioactive source.

In this report, we describe the experimental demonstration of real-time radioactive material detection using a high-power pulsed millimetre-wave source. In particular, we present the measured detection sensitivity and threshold of this method, and compare these values with those calculated using the existing theory. Finally, we discuss the applicability and limitations of the investigated method.

## Results

**Breakdown threshold measurement.** First, we experimentally measured the threshold electric field for plasma breakdown with and without the radioactive material (0.64 mCi $^{60}$Co). Then, we

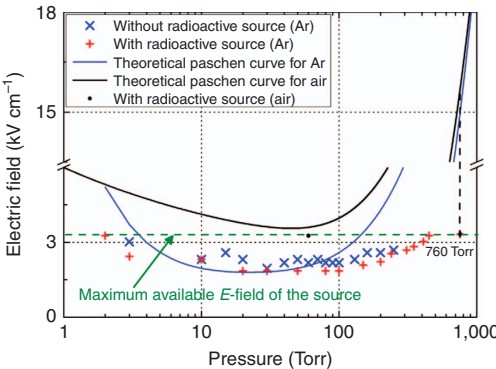

**Figure 1 | Measured threshold fields for breakdown and theoretical Paschen curves for Ar and air.** All of the points (blue crosses, red plus signs and black circles) represent measured data, and the lines (black and blue) represent the theoretical curves. For Ar, measured data are depicted for the cases without radioactive material (blue crosses) and with 0.64 mCi of $^{60}$Co (red plus signs). The measured threshold electric field had $0.15\,kV\,cm^{-1}$ variance. The electromagnetic (EM) wave power was not sufficiently high to induce breakdown in the absence of the external radioactive source at pressures $> 250$ Torr in Ar. However, even though the output power of the gyrotron was insufficient to initiate avalanche ionization, saturation breakdown is observable even at 460 Torr, due to the generation of free electrons by the $^{60}$Co source. The maximum output electric field of the gyrotron, $\sim 3.49\,kV\,cm^{-1}$, is below the required threshold electric field ($\sim 15.7\,kV\,cm^{-1}$ at 760 Torr in Paschen curve) in Ar and air (black dotted vertical line). However, the observed plasma breakdown indicated by the black circles at 60 Torr and 760 Torr in air and 760 Torr in Ar present the possibility of plasma breakdown in the presence of radioactive material at powers lower than those predicted theoretically.

compared the measured values to the theoretical Paschen curves for Ar and air, as illustrated in Fig. 1 (refs 9,10). The maximum available gyrotron power of 32 kW corresponds to an electric field of $3.49\,kV\,cm^{-1}$ (the horizontal dotted green line in Fig. 1). Here, we defined the threshold electric field as the electric field at which 100% breakdown with 20 μs pulse duration of 1 Hz repetition rate occurred in 200 shots experimentally. The distance between the focused spot of the electromagnetic (EM) wave and the radioactive material was set to 20 cm. The measured threshold electric fields without radioactive material are generally in good agreement with the theoretical Paschen curve in the Ar case at pressures between 4 Torr and 100 Torr, the range in which the maximum EM power was above the theoretical threshold breakdown power. In this pressure range, the experimentally measured threshold field in the case in which the $^{60}$Co source was present is $\sim 1.7$ times lower than that without the source[11]. This difference resulted from the generation of a large number of secondary electrons because of the interaction between the gamma rays and the neutral Ar gas molecules, which had low energies ($< 1\,MeV$) due to Compton scattering. Discrepancies between the measured and theoretical values are evident at pressures above 100 Torr[12]. The measured data also show that plasma breakdown occurred at pressures $> 250$ Torr in Ar when the radioactive source was present. Until now, plasma breakdown in an applied electric field significantly lower ($\sim 4$ times lower) than the threshold field has not been reported. However, as shown in Fig. 1, plasma breakdown is observable in the presence of radioactive material due to possible plasma formation in this situation in both Ar and air. Specifically, plasma breakdown is evident at 760 Torr (ambient atmospheric pressure) both in Ar and air with 100% probability of plasma occurrence when the radioactive material was present (no breakdown would be

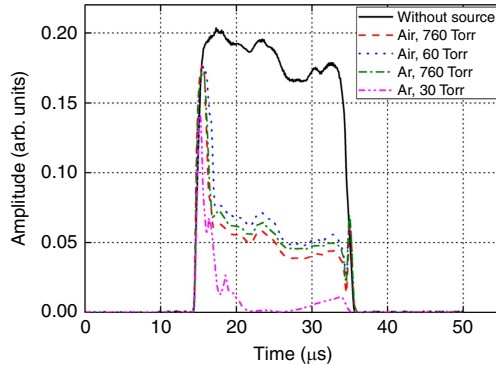

**Figure 2 | RF diode signal waveforms attenuated due to plasma formation in the path of the RF beam.** Typically, the radio-frequency (RF) pulse is turned on at around 15 µs and maintained for 20 µs, as indicated by the black curve, which was obtained without a source. The RF pulses drop quickly because they are attenuated immediately upon plasma formation. The RF signals do not completely drop to zero in air at 60 Torr or 760 Torr or in Ar at 760 Torr, enabling estimation of the plasma density. (For comparison, in Ar at 30 Torr, the RF diode signal represented by the pink dotted line drops almost to zero since the incident RF power is above the threshold power for breakdown, and the plasma density reaches the critical density).

observed at this pressure without radioactive material, of course). Neither the photodiode nor the spectroscopic signal probe picked up the emitted light in the 200–1,000 nm wavelength range due to the extremely low photon intensity; however, the delay time was measurable using the radio-frequency (RF) diode signal when the radioactive material was present. In air, the electric field required for breakdown was $>3.5\,\mathrm{kV\,cm^{-1}}$ throughout the pressure range. However, we observed plasma breakdown in air when the radioactive material was present in under-threshold conditions. For the air breakdown experiment, we performed tests at two representative pressures: 60 Torr and 760 Torr (black circles in Fig. 1), corresponding to the pressure applied in the presence of the minimum threshold field and atmospheric pressure, respectively, and observed the RF pulse shapes in the under-threshold and threshold conditions. Figure 2 depicts the RF pulse shapes obtained when breakdown occurred. The RF pulses were not attenuated completely in the cases in which the incident EM field was much lower than the threshold field, that is, in the under-threshold conditions (60 Torr and 760 Torr in air, and 760 Torr in Ar). Meanwhile, the RF pulse corresponding to 30 Torr in Ar is attenuated almost to zero, indicating that these conditions are the threshold conditions. The density of the formed plasma can be estimated by measuring the RF pulse amplitudes shown in Fig. 2. The transmittance of the EM beam was calculated using a plasma slab model, which is described in detail in Supplementary Note 2. The plasma densities were estimated to be $6.44\times10^{13}\,\mathrm{cm^{-3}}$ at 760 Torr in air, $6.23\times10^{13}\,\mathrm{cm^{-3}}$ at 760 Torr in Ar and $5.87\times10^{13}\,\mathrm{cm^{-3}}$ at 60 Torr in air.

**Ar plasma with radioactive material.** Next, we focus on the Ar gas experiment conducted in the presence of the radioactive material since the gyrotron output power was sufficient to form Ar plasma at pressures between ~2 Torr and 250 Torr, as shown in Fig. 1, which enabled us to study the plasma formation in the presence of the radioactive material in greater detail. A distinct difference between the delay times obtained in the presence and absence of the radioactive material is evident. The probability of no breakdown for the formative delay time

can be expressed as[7,13]

$$P_1(N < n_{cr}, t) = \int_0^{n_{cr}} P(N)\mathrm{d}N = 1 - \exp\left(-\frac{n_{cr}}{\bar{n}}\right), \qquad (1)$$

where $\bar{n}$ (cm$^{-3}$) is the average value of the number of electrons per volume. Further, the electron density approaches the critical density ($n_{cr} \approx 10^{14}\,\mathrm{cm^{-3}}$) at 95 GHz when the plasma frequency is the same as the angular frequency. $P_1(N)$ represents the formative delay time, which is the probability of an avalanche reaching a size corresponding to an electron density $n$.

The probability distribution for the statistical waiting time for the avalanche such that zero electrons are found until per volume up to time, $t$, can be formulated

$$P_2(n = 0, t) = \exp(-St), \qquad (2)$$

where $S$ is the average generation rate of initial electrons owing to the stochastic seeding source in the volume[12]. We assume that the density of free electrons generated by the decay of $^{60}$Co in the breakdown volume is constant and that the source term $S$ is independent of the inner pressure of the chamber, as the primary seed electrons from the radioactive material have high average energies (~0.44 MeV). Hence, the rate of attachment to the molecules can be considered to be the same under all pressures. By combining the two probabilities, that is, those for the formative and statistical delay times, the fitting curve can be obtained by changing the source $S$ for the radioactive material case (black dashed line in Fig. 3) (Supplementary Note 3).

Figure 3 depicts the probability of no breakdown as a function of the delay time. The experiment used to obtain these data was performed using an electric field with an amplitude equal to the threshold value necessary for plasma breakdown at each of the pressures. The black dashed lines indicate the theoretically calculated delay times described above. The delay times represented by the absence of the radioactive material (blue circles) increase as the pressure increases from 30 to 250 Torr. It is evident that the statistical plasma formation delay time distributions extend to longer times in the absence of the radioactive source. The measured minimum delay time in the presence of the radioactive material is $<2.4\,\mu s$ for each of the pressures, as shown in Fig. 3. At 250 Torr, not only the statistical delay time, but also the formative delay time, is reduced significantly in the presence of the radioactive source. These characteristics are strong indicators of the presence of a radioactive source nearby.

With increased confidence that the existence of radioactive material significantly changes the delay time before breakdown, we evaluated the ability of the proposed method to indicate the presence of radioactive material in real time. A remotely controllable gate was mounted on the chamber containing the radioactive material. We monitored the delay time before plasma breakdown for each EM pulse and performed real-time measurements as a function of time with (gate open) and without (gate closed) the radioactive source, as well as with different distances between the radioactive source and the EM beam focal point. Figure 4a shows that the gyrotron's ability to indicate the presence of radioactive material in real time is strong. The delay time is markedly shorter in the presence of the radioactive material; thus, it could be detected in real time. Comparison of the data obtained with (red circles) and without (blue squares) the radiation source indicates that the delay time distribution in the ordinary state is significantly wider than that in the presence of the radioactive source, since the free electrons present in this case have a more random distribution in time. In addition, the average delay time and the width of the delay time distribution both increase in the presence of radioactive material as the distance increases from 20 to 120 cm (given the limitations of the laboratory setup, the distance could be at most 120 cm), as

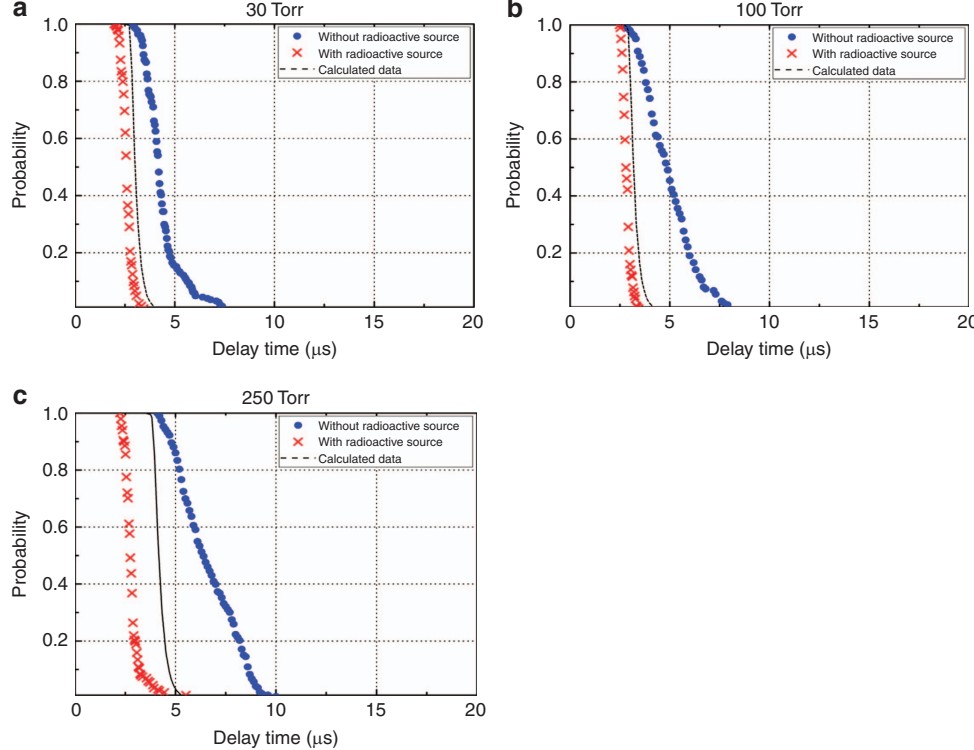

**Figure 3 | Probability of no breakdown as a function of the delay time.** (**a**–**c**) At each pressure, the delay time was measured when the threshold RF electric field was applied. The threshold RF electric field amplitude was defined as the applied RF electric field amplitude at which 100% plasma breakdown with 20 μs pulse duration occurred over 200 shots at a repetition rate of 1 Hz. The measured electric field variance was 0.15 kV cm$^{-1}$. The red crosses and blue circles indicate the experimental data obtained with and without the radioactive source, respectively. The theoretical curves (black dashed line) were calculated using the Laue plots in combination with the formative and statistical waiting times, as described in Supplementary Note 3 (refs 7,13,14).

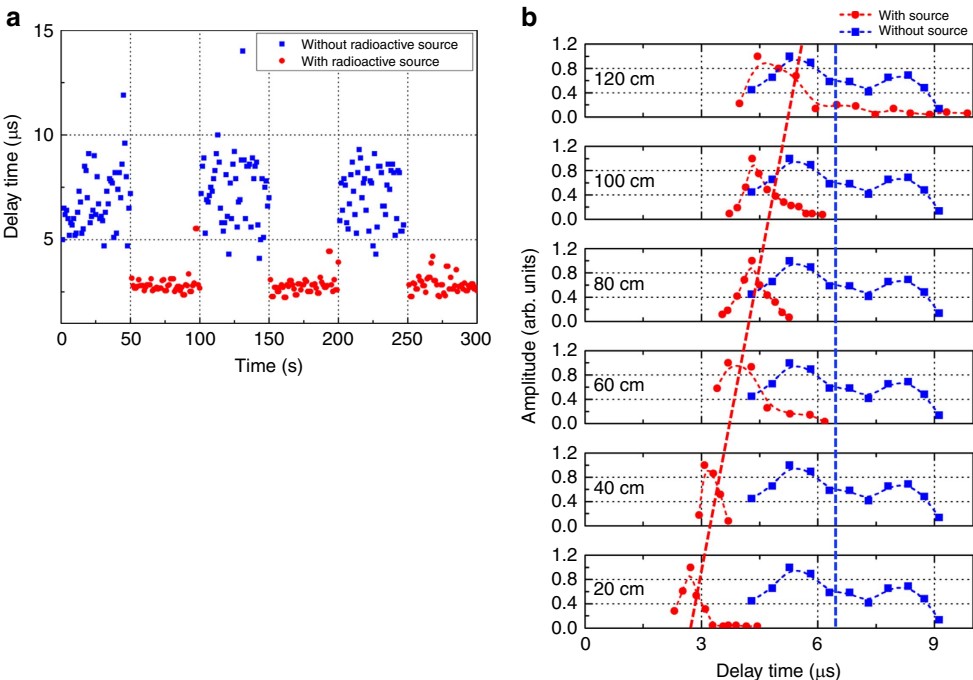

**Figure 4 | Experimentally determined real-time ability to detect the presence of radioactive material.** The experiment was implemented at 19 kW and 250 Torr. (**a**) Real-time measurement of the total delay time with and without the radioactive source. The source was in a lead box 20 cm from the focal point, and the lead box was opened and closed every 50 shots by an autocontrolled gate. The minimum delay times required for discharge with and without the source were ~2.2 μs and 4.1 μs, respectively. (**b**) Delay time distributions measured with the $^{60}$Co source 20–120 cm from the EM beam focal point. The delay time distributions obtained with and without the radioactive source are distinct from one another at each of the distances.

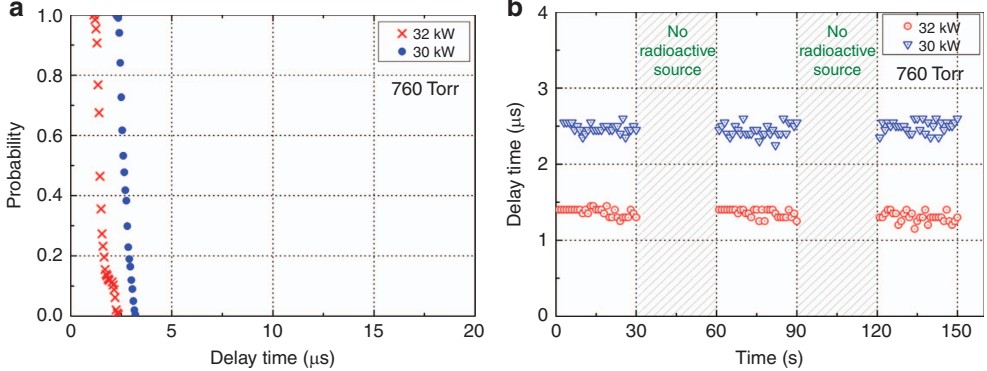

**Figure 5 | Experimental delay time measurements in the presence of radioactive material in Ar gas.** (**a**) Probability of no breakdown versus delay time. (**b**) Real-time measurements of delay time variations at output powers of 30 and 32 kW. No breakdown is observed in the absence of radioactive material, as indicated by the grey hatched regions.

shown in Fig. 4b. The centres of the delay time distributions obtained with the radioactive material present, which are indicated by the red dotted line, shift to the right and become closer to those of the distributions obtained without radioactive material, which are indicated by the blue dotted line in Fig. 4b. The delay time distributions over 200 shots with and without the radioactive material are distinctly separated when the detector is close to the source. Although they start to overlap when the distance is 120 cm, they are still discernable, which demonstrates the sensitivity of the measurement technique. As shown in Fig. 4b, the delay times with and without radioactive material can be distinguished from one another up to ∼120 cm.

**Detection sensitivity.** Experimentally, we could detect 0.5 µg (equivalent to 0.64 mCi activity) of $^{60}$Co up to ∼120 cm away from the focus of the incident EM wave, as shown in Fig. 4b. Now, we compare these experimentally obtained detection sensitivities to the theoretical results calculated based on plasma on/off phenomena. The theory assumes that the incident EM beam is focused and that there is rarely a free electron present in the breakdown-prone volume in the absence of radioactive material within the pulse duration of the incident EM wave, resulting in close to zero probability of breakdown. The theoretically detectable mass ($M(g)$) of $^{60}$Co can be calculated using the following previously reported formula[14]:

$$M(g) \geq \frac{4\pi L_{\gamma,a} R^2}{A V \tau_{THz}} \frac{\Delta E}{\langle E \rangle} \frac{v_i}{v_{i,eff}} \exp(d/L_\gamma), \qquad (3)$$

where $L_{\gamma,a} = 1/n_a \sigma_a$ ( $= 280$ m, $n_a$ is the density of air) is the range of gamma ray propagation and is determined by the average absorption cross-section, $\sigma_a$; $d$ is the distance from the radioactive source to the focused beam spot; $A$ is the specific activity, which is $1.1 \times 10^3$ Ci g$^{-1}$ for $^{60}$Co; $V = 0.17$ cm$^3$ is the breakdown-prone volume corresponding to $\hat{P} = P/P_{th}$, the ratio of the wave power to the threshold power in the focal plane; $\tau_{THz}$ is the pulse duration (s); $\frac{\Delta E}{\langle E \rangle}$ is the ratio of the energy required to produce one secondary electron–ion pair to that of the primary electrons generated by the gamma rays; $\frac{v_i}{v_{i,eff}}$ is the ratio of the ionization frequency to the effective ionization (net ionization) frequency; and $L_\gamma = 1/n_a \sigma_T$ ( $= 130$ m) is the gamma ray propagation distance and is determined by the total interaction cross-section, $\sigma_T$, in air. Using equation (3), the detectable masses of $^{60}$Co at 20 and 120 cm were calculated to be 0.067 mg and 2.4 mg, respectively. Considering that there are distinct differences between the delay times with and without

radioactive material that were measured at distances of 20–60 cm (Fig. 4b), the experimentally detectable mass of 0.5 µg corresponds to a sensitivity at least 130 times greater than the theoretical sensitivity determined based on plasma on/off phenomena. The delay times with and without radioactive material those were measured at 120 cm still exhibit noticeable differences (Fig. 4b), in which case the sensitivity increases to 4,800 times the theoretical sensitivity. The discrepancies between the experimentally determined and theoretically predicted minimum detectable masses are attributed to the fact that our experiment was based on using plasma delay time measurements to discern the presence of the radioactive material, whereas the theory was based on plasma on/off phenomena. Thus, plasma breakdown delay time measurements result in more sensitive detection than does reliance on plasma on/off phenomena. Our increased sensitivity may also relax the frequency requirement of the millimetre/THz-wave source.

**Under-threshold breakdown with radioactivity in Ar and air.** In the previous subsections, we discussed the breakdown of Ar in the presence of radioactive material. An air breakdown study with radioactive material at atmospheric pressure was also necessary to prove the effectiveness of the proposed idea in a real environment. However, the maximum possible electric field intensity for the gyrotron was lower than that according to the theoretical Paschen curve of air, as shown in Fig. 1. Nonetheless, very interestingly, we experimentally observed under-threshold breakdowns, that is, breakdowns that occurred in applied electric fields much weaker (more than four times weaker) than the threshold field necessary for plasma formation in the vicinity of the radioactive material. Of course, no plasma breakdown occurred in the absence of radioactive material in this electric field.

First, we investigated the under-threshold breakdown with radioactive material in Ar at 760 Torr. As shown in Fig. 5a, the delay time is not widely distributed as it was in Fig. 3 (the delay times are not statistically distributed) since the existence of radioactive material eliminates the statistical nature of the delay time distribution. The delay time decreases significantly as the output power increases. Figure 5b demonstrates the feasibility of using the proposed method for real-time radioactive material detection and shows that the delay time is dependent upon the EM power. The delay times were measured in real time with and without the radioactive material as the incident EM power was varied. No breakdown was observed at either 30 or 32 kW when the radioactive material was absent, as

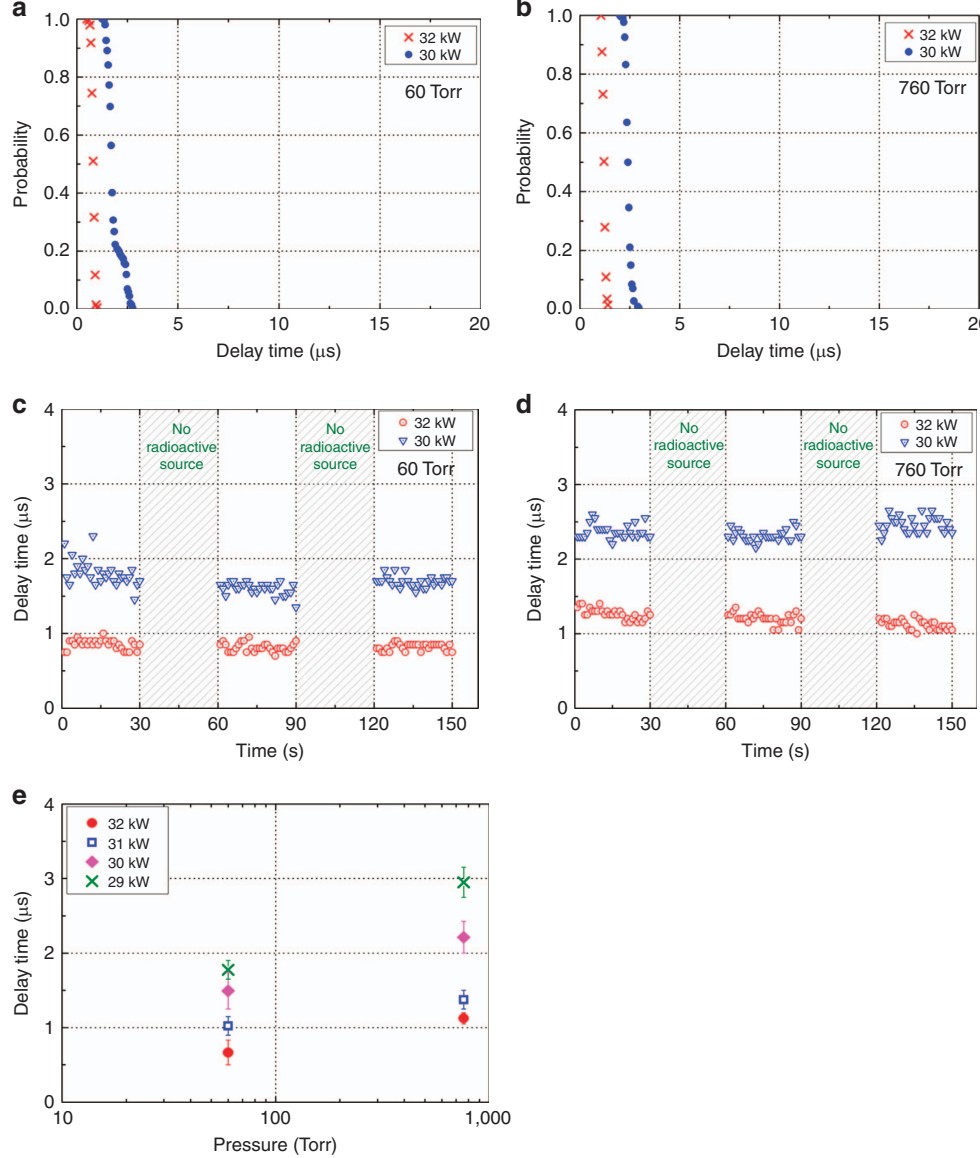

**Figure 6 | Delay times experimentally measured in the presence of radioactive material in Air.** (**a,b**) Probability of no breakdown versus delay time at 60 Torr and 760 Torr, respectively. (**c,d**) Real-time delay time measurements obtained at 60 Torr and 760 Torr, respectively, along with 50% of the cumulative data measured with respect to the initial delay time. A gate on the lead box enclosing the radioactive source was opened and closed every 30 s by an autocontrolled gate. (**e**) Dependence of delay time on pressure at various EM powers.

indicated by the hatched regions in Fig. 5b. Clear differences are evident between Figs 4a and 5b; both cases indicate that the presence of the radioactive material caused the statistical nature of the delay time distributions to disappear in real time, but under-threshold breakdown (Fig. 5b) is impossible without exposure to radioactive material.

The theoretical Paschen curve (Fig. 1) predicts that almost the same threshold electric field is required for plasma breakdown at atmospheric pressure in air and in Ar when there is no radioactive material[12]. However, with radioactive material nearby, the threshold electric field is significantly reduced, according to the Ar gas experiment. The experimental results obtained using Ar gas suggest that plasma breakdown may be possible in air at 760 Torr, even if the incident EM power is much lower than the threshold power. Therefore, we performed the air breakdown experiment in the presence of the radioactive material. The output power of the gyrotron was varied between 29 and

32 kW at 60 Torr and 760 Torr in air. According to the Paschen curve, 60 Torr is the minimum threshold pressure for pressurized air breakdown (Fig. 1). The distance between the radioactive source and the focal point of the incident EM beam was set to 20 cm for the experiment, which was performed in ambient air at a temperature of 23 °C ( ± 0.5 °C) and a humidity of 62% ( ± 2%).

We successfully observed plasma breakdown in air at 60 Torr and 760 Torr in the presence of 0.64 mCi of $^{60}$Co with incident EM powers of $\sim$30 kW, that is, around ten times smaller than the power required for plasma breakdown without radioactive material. Figure 6a,b depict the probabilities of no breakdown at different incident EM powers in the presence of radioactive material at 60 Torr and 760 Torr, respectively. The breakdown delay times shown in Fig. 6c,d were measured in real time, similarly to in the Ar experiment. Breakdown cannot occur without radioactive material, as indicated by the hatched regions

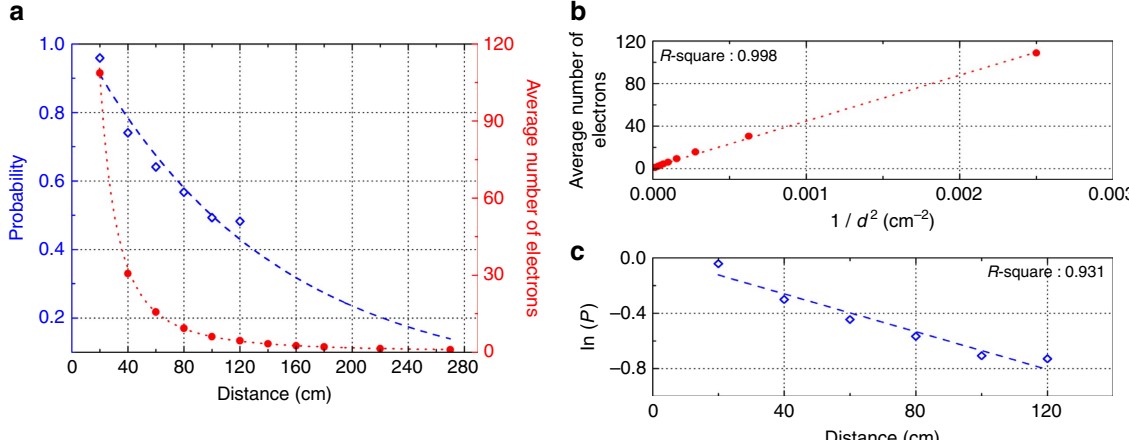

**Figure 7 | The probability of plasma breakdown and MCNPX results.** Experimentally determined plasma breakdown probability and MCNPX-simulation-based average number of free electrons as functions of distance. Under normal conditions (that is, in the absence of the radioactive source), plasma breakdown is not observable for an incident power of 28 kW in Ar at 400 Torr. (**a**) Plasma formation probability, which decreases exponentially with increasing distance, and average number of free electrons, which decreases in proportion to $1/d^2$. The circles and diamonds represent the MCNPX simulation results and the experimentally measured data, respectively, and the dotted curves are the fit curves. (**b**) Linear fit of the average number of electrons to $1/d^2$. (**c**) Linear fit of the logarithm of the probability to $d$. The linear fits are in excellent agreement with the experimental data. The $R^2$ values represent how close the data are to the regression lines.

in Fig. 6c,d. The measured delay times as well as the dispersions of the delay times in Ar and air at 760 are in close agreement with one another (Fig. 5b for Ar at 760 Torr versus Fig. 6d for air at 760 Torr). These results represent the clear experimental demonstration of the real-time monitoring of radioactive material in ambient air by using a low EM power ($\sim 30$ kW) to initiate the plasma. We also observed the dependence of the delay time on the incident power, as shown in Fig. 6e. As the pressure increases, the delay time increases accordingly due to the increase in the power required for plasma breakdown. Also, the delay time becomes shorter as the incident EM power increases at a fixed pressure.

The considerable reduction observed in the measured threshold electric field ($\sim 3.5$ kV cm$^{-1}$ during the experiment performed in the presence of radioactive material versus $\sim 16$ kV cm$^{-1}$ in theory in the absence of radioactive material) necessary to initiate plasma breakdown in air at atmospheric pressure in the presence of radioactive material requires further analysis. We postulate that the increased conductivity in the breakdown-prone volume leads to the reduction in the electric field amplitude necessary for breakdown. We introduce a field-reduction factor, $\beta$, to express the reduced electric field required for breakdown owing to the presence of radioactive material:

$$\beta E_0 = E_{cr}, \qquad (4)$$

where $\frac{1}{\beta} = \ln(\frac{n_{cr}}{n_0}) / \ln(\frac{n_{cr}}{n_0^*}) = \ln(\frac{n_0^*}{n_0})$ (Supplementary Note 4). Here, $n_0$ is the seed electron density when there is no radioactive material; for typical atmospheric cases, a seed electron number density of 1–10 cm$^{-3}$ is widely accepted. $n_0^*$ is the seed electron density in the presence of radioactive material, and $n_{cr}$ is the critical plasma density at 95 GHz. Because the radioactive material generates high-energy gamma photons, high-energy electrons are produced in turn. The initial high-energy electrons with a density of $n_{0e}$ produce secondary knock-on electrons via collisions with the molecules present in the breakdown-prone volume. To produce one secondary electron–ion pair, an average energy of $\sim 34$ eV is needed[11]. The energy distribution of the gamma photons and high-energy electrons of 0.64 mCi $^{60}$Co at 20 cm from the source can be determined using Monte Carlo N-Particle eXtended (MCNPX ver. 2.50) code (Supplementary Fig. 2). The calculated number of secondary knock-on

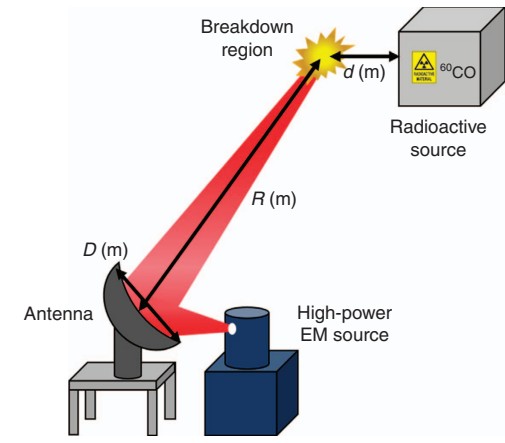

**Figure 8 | Schematic of a possible setup for the detection of radioactive material inside a container.** The distance from the high-power EM source to the breakdown point is $R(m) = 2D^2/\lambda$, where $D$ is the size of the antenna's aperture, and $\lambda$ is the wavelength of the incident beam.

electrons produced by a single high-energy electron is $\sim 12{,}600$ (Supplementary Note 4). Considering the pulse length of the applied RF field ($\sim 1\,\mu$s) before the formation of the plasma, the total secondary knock-on electron number density attributable to the high-energy electrons is $\sim 1.3 \times 10^8$ cm$^{-3}$ (Supplementary Note 4). The significant increase in the number of background-free electrons owing to the high-energy electrons generated by the radioactive material results in an increase in the conductivity of the RF-focused volume, resulting in breakdown in the area. The field-reduction factor increases gradually as the number of background-free electrons increases (Supplementary Fig. 3). For an initial seed electron number density of $1.3 \times 10^8$ cm$^{-3}$, the calculated value of $\beta$ is $\sim 2.5$, which means that the required electric field is reduced by a factor of 2.5 as compared to the field required when a radioactive material is not present. On the basis of this analysis, the reduction in the electric field required for breakdown (experimentally determined reduction factor of 4 versus theoretical reduction factor of 2.5) can be

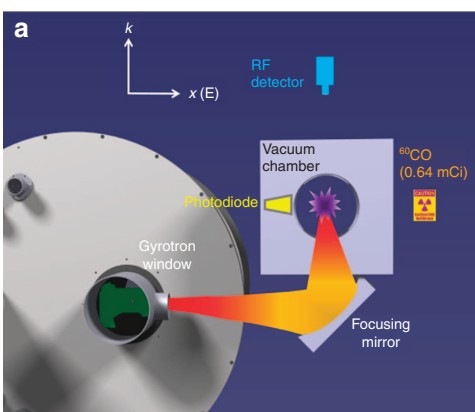
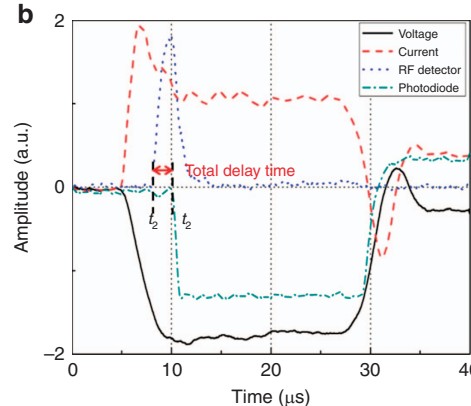

**Figure 9 | Experimental layout for the remote detection of a radioactive source.** (**a**) The setup consists of a millimetre-wave power source, a gyrotron, a spherical focusing mirror 42 cm from the gyrotron window to focus the beam inside the vacuum chamber, an RF detector 60 cm from the centre of the vacuum chamber, a photodiode, and a 0.64 mCi sample of $^{60}$Co as the radioactive source. The EM wave propagates from the gyrotron window and is focused at a small spot inside the vacuum chamber, as indicated in red. At the focal point, the electric field is polarized in the $x$-direction and propagates in the $k$-direction, as defined in the figure. The formed plasma travels in the $k$-direction. (**b**) Typical oscilloscope signals (voltage, current, RF detector and photodiode signals). The gyrotron is operated at a voltage and current of 40 kV and of 7 A, respectively.

understood to a certain extent, although a slight discrepancy still exists between the experimental and theoretical observations. The additional reduction observed experimentally may be due to uncontrollable experimental conditions, including the fact that the exact molecular contents are unknown and an aerosol may be present[11,12].

**Sensitivity and detection range limitation.** Finally, we determined the detection range of the setup by measuring the plasma breakdown probability as a function of the distance. For these measurements, the incident gyrotron beam was irradiated 300 times with a repetition rate of 1 Hz at each pressure. The measurements were performed at a pressure of 400 Torr in Ar with an incident EM wave power of 28 kW, in which case plasma breakdown did not occur in the absence of the radioactive material. Figure 7a shows that the probability of a plasma avalanche occurring decreases $d$ increases. To investigate this phenomenon, we simulated the generation of free electrons by applying the same conditions that were employed in our experiment and using the MCNPX code (data denoted by red circles in Fig. 7a). The average number of free electrons in a volume of 1 cm³ decreases with increasing distance, with the number eventually decreasing to 1 electron cm⁻³ at a distance of 270 cm from the beam focal point; this value is approximately the same as the number of background electrons. In the short-distance ($d \ll L_\gamma$) approximation, the photon density decreases with $1/d^2$ (ref. 14). The simulated average numbers of free electrons that are presented in Fig. 7b closely follow the expected trend. In addition, the exponential decay of the probability as a function of distance was observed, and the results are shown in Fig. 7c. The probability of plasma breakdown becomes < 20% at distances > 220 cm, at which the average number of free electrons is < 2.

In an actual situation, the incident EM wave can be emitted from far away (at a distance $R$) by using an antenna, as shown in Fig. 8. The location of breakdown due to the radioactive source can be adjusted to increase the detection sensitivity. At 95 GHz, an antenna size of 1.2 m enables the detection range to be 1 km from the source. However, air turbulence may limit the maximum focal location. Let us consider the effect of beam wander on the detection range[15]. For the weak turbulence case, the detection range $R$ is

$$[R(m)]^3 < \frac{10^{-2}[\rho_0(m)]^2}{0.434 C_n^2 K},\qquad(5)$$

where $\rho_0$ is the beam width; $C_n^2$ is the strength of turbulent refractive-index irregularities and is between $3 \times 10^{-13}$ and $6 \times 10^{-17}\,\mathrm{m}^{-2/3}$, which correspond to strong and weak turbulence, respectively; and $K$ is an integral coefficient $(\mathrm{m}^{-1/3})$ defined in ref. 15. Under these conditions, $R$ varies from 50 m to 1 km depending on the strength of the turbulent parameter. Therefore, the detection range of the proposed method is limited according to the air turbulence strength.

## Discussion

The results of this experimental study confirm that it is possible to detect radioactive material remotely using a high-power millimetre-range gyrotron. We could observe the elimination of the statistical nature of the delay time distribution in the presence of radioactive material by using a 95 GHz, 30 kW gyrotron source. Owing to the plasma breakdown delay time measurements, the sensitivity of the proposed method in terms of the detectable mass was much higher (at least 130 times) than that predicted theoretically based on plasma on/off phenomena. The dependences of the delay time for plasma breakdown on the applied RF power and pressure were evaluated in Ar and air. The real-time remote detection of radioactive material was achieved in Ar and air at pressures of up to 760 Torr through delay time measurements. Furthermore, we observed that the presence of radioactive material greatly reduced the breakdown threshold electric field in the atmosphere; a quantitative analysis was performed to explain this observation by introducing an electric field-reduction factor. Compared to the currently available radioactive material detection technologies, the proposed technique allows for the detection of radioactive material at greater distances from the detector; detection at distances as large as 1 km should be feasible using an antenna of the proper size if the atmospheric turbulence is not strong.

## Methods

A schematic of the experimental setup is shown in Fig. 9a. We used a laboratory-made 95 GHz gyrotron oscillator with a pulse duration of 20 μs (refs 16,17). The radius of the single-pulse beam, which was focused at a point inside the vacuum

chamber, was ~5 mm. The repetition rate of the gyrotron was 1 Hz, which enabled us to obtain breakdown statistics over shots. The generated plasma travelled in the direction opposite to the propagation direction of the EM wave. The transmitted EM wave was detected by a RF detector to determine the RF onset time $t_1$, and the time $t_2$ at which the RF pulse was attenuated due to plasma formation. The photodiode signal was also used to monitor the onset time ($t_2$) of the fluorescent light generated by the formation of the plasma. The interval between $t_1$ and $t_2$ represents the total delay time and consists of the plasma formation time and the statistical delay time (Fig. 9b). The formative delay time is the time taken for the initial seed electron density to reach the critical plasma density at the operating frequency. The statistical delay time represents the period before the appearance of the initial seed electron. A 0.64 mCi sample of $^{60}$Co (creating gamma rays with energies of 1.173 and 1.332 MeV through decay) was initially placed 20 cm from the focal point of the incident EM wave; this distance was varied up to 120 cm. The maximum available output power of the gyrotron was 32 kW. The power was varied to induce plasma breakdown under different pressure conditions (the pressure was varied from 2 to 760 Torr). We performed an argon (Ar) gas experiment first since the theoretical threshold EM power for breakdown in Ar gas is lower than the maximum gyrotron power in some pressure ranges, which enabled us to study the breakdown statistics in the presence of radioactive material. In addition, the air breakdown experiment was performed only with the radioactive source, even though the amplitude of the incident electric field was lower than the minimum required amplitude.

**Data availability.** The data that support the findings of this study are available from the corresponding author on reasonable request.

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

## Acknowledgements

This research was supported by the National Research & Development Programme (NRF-2013R1A1A2061062) of the National Research Foundation of Korea (NRF), which is funded by the Ministry of Science, ICT & Future Planning, Korea, by Basic Science Research Program through the National Research Foundation of Korea (NRF) funded by the Ministry of Science, ICT & Future Planning (2017R1A2B2003689), and by the NRF Grant of Korea funded by the Ministry of Education (NRF-2012-Global PhD Fellowship Programme). We would like to thank Ms Hyejin Lee at UNIST UCRF for her assistance with handling the radioactive material during the experiment and Dr Evgenya Simakov at Los Alamos National Laboratory (LANL) for her comments on the manuscript.

## Author contributions

E.M.C. conceived and supervised the project. D.K. designed the experiment, and D.K., D.Y. and A.S. performed the plasma breakdown experiments. M.S.C., I.L. and S.G.K. assisted in the experimental setup. D.K. and A.S. performed the simulations for the theoretical calculation of delay time. D.K. and E.M.C. analysed the observed plasma breakdown condition, and D.Y. and A.S. participated the analysis. D.K. and E.M.C. wrote the manuscript. All authors provided experimental support and participated in discussion.

## Additional information

**Competing interests:** The authors declare no competing financial interests.

