## [Peer Review File · Nature Communications]

Reviewers' comments:

Reviewer #1 (Remarks to the Author):

This MS is devoted to a very interesting and important issue of remote detection of radioactive materials by using high-power electromagnetic (EM) radiation. It can be published after a major revision.

My main concern is motivated by the feeling that in some places this MS looks like a commercial, rather than a serious scientific paper.

To illustrate my point, I will start from the abstract as an example. This abstract contains such statement like: "the detection range of the method can be extended to the kilometer range" or "the detection sensitivity of the method is 1000 times higher than that suggested by theoretical calculations". (The same statements are repeated in Conclusions.) Of course, the range can be extended to one kilometer and, if you develop a multi-MW gyrotron, it will be many-km range and so on, but from the measurements done in about 1 meter proximity to the source I wouldn't draw so strong conclusions.

Now, I will move along the MS with comments:

1) Page 2 (line 44): the proposal [2, 3] to use a laser-induced plasma for remote detection of radioactive materials is also based on the reduction of the delay time, that is the key point of the present paper.

2) Page 3 (line 61): it is unclear what is unrealistic in the estimates done in Ref. [10].

3) Page 4 (line 79): this critical density depends on the wave frequency. It should be mentioned what frequency corresponds to this density. On lines 86-87 again the same unmotivated statement about "1000 times" is repeated.

4) Page 5 (line 99). It is critically important for further estimates to mention what is the breakdown-prone volume for a focused wave beam of the given wavelength and power for the pressures considered below.

5) Page 7: Eq. (4) (mistakenly numbered as (3)) and the text below. Again, what is the breakdown-prone volume (I got an impression that the authors simply do not understand what means this V in Eq. (4) because they call it on line 164 " V is the volume of the incident wave". Next, in the ratio of energies, they call ΔE in the numerator "the energy of the secondary electrons", while this is the energy required for producing one secondary electron-ion pair. The importance of accurate definition of the breakdown-prone volume can be illustrated by a simple example: I assumed that this volume is equal to 0.1 cubic centimeter and got for a 20 microsecond pulse the detectable mass equal to 18 microgram. This value exceeds their detectable level by less than 40 times only (instead of 1000!)

6) Page 8 (lines 172-3): I recommend the authors to change their point of view on comparison of their method with the existing one. The method proposed by other authors is the first one and, as such, it proposes the simplest way to detect the presence of concealed radioactive materials (viz. the breakdown rate which depends on radioactive gammas). The authors propose a more delicate method based on measuring the difference in the delay time (essentially, the same as the authors of [2, 3]). Yes, this method has some advantages, but it also requires more accurate registration technique - and this is what should be explained.

7) Page 8 (line 175): I don't understand where the authors took the requirement to have the volume size less than 1 cubic mm: all what was required in previous studies is to have it less than 0.1 cubic cm, i.e. two orders more!

8) Page 9 (line 206). What is so surprising in the fact that the probability of the breakdown decreases as the distance from the radioactive source increases? In Ref. [10], Eq. (9) shows that the density of gammas ionizing air falls with the distance from the source faster than $1/R^2$. So, the question could be only: to what extent results shown in Fig. 6 of the MS agree with this dependence? This statement is also valid for the text on lines 234-5 about scalability of the results.

Reviewer #2 (Remarks to the Author):

The paper presents results of interesting experiments demonstrated how sensitive could be a remote detection of radioactive materials when using high power microwave radiation. These results could be interesting for readers. Unfortunately the paper text is written in too unclear and inaccurate manner. Some particular remarks are listed below.

1. The equation (3) is not justified at all. Moreover it seems to be wrong because it is quite

different from the rigorous mathematical solution of the similar problem given in Ref.. 8.

2. A quantity $n(t)$ which is given in equation (2) is termed as averaged electron density but the quantity $n(t)$ is nondimensional. It should be noted that the quantity $n(t)$ is also presented as non-dimensional in Eq(1).

3. The quantity n_{cr} which is given in Eq(3) is also nondimensional and therefore it cannot be treated as critical plasma density which is measured in $1/\text{cm}^3$.

4. It is absolutely unclear how the theoretical line was obtained in Fig. 3.

5. A caption to Fig. 3 states that experimental results are obtained taking electric field amplitude close to the breakdown threshold. This is absolutely not sufficient because these results are very sensitive to a difference between the applied field amplitude and its breakdown value.

6. It is not clear what value the authors treats as the breakdown threshold which is shown in Fig. 5.

7. The authors state several times that the radioactive source create some electron density which value decreases with an increase in the distance from the source. However this can be said so only when speaking about average electron density. The actual electron density fluctuates in time and these fluctuations plays the main role in the statistical fluctuations in the breakdown delay time which is studied in experiment.

8. The authors predict very high sensitivity of their detection method in air based on the experimental results obtained in argon. But the breakdown process in air and argon are quite different and therefore the quantitative conclusion of authors seems to be incorrect.

The above list is not complete but to my mind it is quite sufficient to reject a publication of the paper in present form.

Reviewer #3 (Remarks to the Author):

1. This experimental study was motivated by the concept of remote detection of concealed radioactive materials by a focused terahertz (THz) radiation proposed by [V. L. Granatstein and G. S. Nusinovich, J. Appl. Phys. 108, 063304 (2010)].

2. According to this concept, a high-power THz radiation should be focused in a small spot where the field intensity exceeds the breakdown threshold. However, this volume should be so small that in the absence of ionizing sources in its vicinity the probability to have there any free electrons is low.

3. The current study was conducted by focusing of 95 GHz beam in Ar (which has a distinctly different chemistry than air) at almost diffraction limited, but even then only at pressures smaller than atmospheric.

4. The two plots that present the main results are intriguing to the experts, I wish they had been placed in a better-written paper.

5. No comparison with realistic air conditions (including impurities and weakly tight electrons) is supplied. Thus, the claim for the possible remote application is not supported by the current data and no discussion of how this data can be used in more realistic environment is presented

6. Moreover, for remote applications the effect of the atmospheric turbulence on propagation and focusing of sub-THz wave beams in air should be considered. Even under very weak atmospheric turbulence, the effect of the turbulence is small only for distances up to 20m (1). At higher distances the required power will be well beyond the current technology.

7. In conclusion: This work can be of inherent interest to specialists from the remote detection of radioactive materials community and should be published in some forum. However, it not is a sufficiently strong case for publication in Nature Comm.

1. Nusinovich et al J. Appl. Phys. 113, 233303 (2013)

Reviewer #1 (Remarks to the Author):

This MS is devoted to a very interesting and important issue of remote detection of radioactive materials by using high-power electromagnetic (EM) radiation. It can be published after a major revision. My main concern is motivated by the feeling that in some places this MS looks like a commercial, rather than a serious scientific paper. To illustrate my point, I will start from the abstract as an example. This abstract contains such statement like: "the detection range of the method can be extended to the kilometer range" or "the detection sensitivity of the method is 1000 times higher than that suggested by theoretical calculations". (The same statements are repeated in Conclusions.) Of course, the range can be extended to one kilometer and, if you develop a multi-MW gyrotron, it will be many-km range and so on, but from the measurements done in about 1 meter proximity to the source I wouldn't draw so strong conclusions.

Now, I will move along the MS with comments:

Comment 1: 1) Page 2 (line 44): the proposal [2, 3] to use a laser-induced plasma for remote detection of radioactive materials is also based on the reduction of the delay time, that is the key point of the present paper.

Our response: Thank you for the comment. Reference [2] describes the frequency modulation of the RF probe beam in the presence of radioactive material, and reference [3] indeed provides theoretical analysis of the delay time change due to the presence of radioactive material. We have added a new sentence as suggested by the reviewer (revised manuscript page 2, lines 21–23), which is also provided below in red font. Reference [3] states that the researchers found that there was a relationship between the amount of radioactive material and the delay time. According to their theoretical analysis, the delay time variation was ~1 ns when the pumping laser beam intensity and the radiation enhancement factor increased by factors of 10. On the other hand, our technique does not require an additional pumping laser, and we present actual experimental observations of the shortening of the delay time in the presence of radioactive material.

Page 2, lines 21–23: “For instance, a laser-induced plasma was proposed for use in the remote detection of radioactive material based on the reduction of the delay time before plasma formation.^{2,3,,}”

Comment 2: 2) Page 3 (line 61): it is unclear what is unrealistic in the estimates done in Ref. [10].

Our response: Thank you for mentioning this point. Based on our calculations, the detectable mass of 2.4 mg of ⁶⁰Co corresponds to 3.03 Ci. This amount was converted into the radiation dose rate using the equation (see Supplementary Note 1)

$$D \text{ (Sv)} = \frac{1.32 \cdot S \cdot d^{-2}}{R}$$
where 1.32 R· $\frac{\text{m}^2}{\text{Ci}\cdot\text{h}}$ is the gamma constant for ⁶⁰Co, $R = 0.01$ Sieverts (Sv), S is the radioactivity (Ci), and d is the distance from the source (m). After performing this calculation, the value of 3.03 Ci at 1.2 m was converted into 27.7 mSv/h (the radiological limit is 1 mSv/y

for the general public). Since this level of radioactivity is quite high, we expect that a commercial Geiger–Muller detector could detect radioactive material at a great distance (~tens of meters). Therefore, it might be not significantly beneficial to use the plasma breakdown method with electromagnetic waves. However, if radioactive material is smuggled in cargo with shielding that is sufficient to conceal it, the commercially available detectors would not be suitable to sense it. Since the sentence in the original manuscript could be misread, it has been removed.

Comment 3: 3) Page 4 (line 79): this critical density depends on the wave frequency. It should be mentioned what frequency corresponds to this density. On lines 86-87 again the same unmotivated statement about "1000 times" is repeated.

Our response: Thank you for the comment. We have included the wave frequency in the revised manuscript (page 9, lines 2–4) as the reviewer suggested by modifying the corresponding sentence, which is shown below in red font. As the reviewer noted, the statement “1000 times” was unmotivated; thus, it has been removed.

Page 9, lines 2–4: “... and the electron density approaches the critical density ($n_{cr} \approx 10^{14}/\text{cm}^3$) at 95 GHz when the plasma frequency is the same as the angular frequency.”

Comment 4: 4) Page 5 (line 99). It is critically important for further estimates to mention what is the breakdown-prone volume for a focused wave beam of the given wavelength and power for the pressures considered below.

Our response: We appreciate your comment. Let us clarify the definition and the calculation of the breakdown-prone volume in our experiment. We defined the breakdown-prone volume as the volume of the incident wave beam whose power exceeded the threshold power required for breakdown so that breakdown due to the RF beam would be possible.⁴ We calculated the breakdown-prone volume using the equations from reference [4], as shown in the manuscript. The size of the beam in a focal plane can be expressed as

$$\rho_0 = \frac{1}{\sqrt{\pi}} \cdot \frac{L \lambda}{R},$$

where L is for the distance from an antenna to the focal point of the beam, R is the radius of the antenna, and λ is the wavelength of the RF beam. In our experiment, ρ_0 was 0.32 cm. From this value, we finally obtained the breakdown-prone volume using⁴

$$V = \frac{\pi}{3} \cdot \frac{\rho_0^4}{\lambda} \left\{ \left[5 + \frac{\hat{P}}{P} \sqrt{P-1} - 4 \arctan \left(\sqrt{\frac{\hat{P}}{P-1}} \right) \right]^2 \right\},$$

where $\frac{\hat{P}}{P_{th}}$ is the ratio of the wave power to its threshold value in the focal plane. For $\frac{\hat{P}}{P_{th}} = 2$, the breakdown-prone volume in our experimental case was calculated to be 0.17 cm³.

As suggested by the reviewer, the following clear definition of the breakdown-prone volume has been added in the revised paper:

Page 12, lines 18–20: “... $V = 0.17 \text{ cm}^3$ is the breakdown-prone volume corresponding to

$\hat{P} = P / P_{th}$, the ratio of the wave power to the threshold power in the focal plane ...”

Comment 5: 5) Page 7: Eq. (4) (mistakenly numbered as (3)) and the text below. Again, what is the breakdown-prone volume (I got an impression that the authors simply do not understand what means this V in Eq. (4) because they call it on line 164 "V is the volume of the incident wave". Next, in the ratio of energies, they call delta E in the numerator "the energy of the secondary electrons", while this is the energy required for producing one secondary electron-ion pair. The importance of accurate definition of the breakdown-prone volume can be illustrated by a simple example: I assumed that this volume is equal to 0.1 cubic centimeter and got for a 20 microsecond pulse the detectable mass equal to 18 microgram. This value exceeds their detectable level by less than 40 times only (instead of 1000!)

Our response: Thank you for the comment. As mentioned in our previous response, the definition of the breakdown-prone volume has been clarified (page 12, lines 12–13), as shown below. Also, equation (3) (in the original manuscript) has been revised to correct the numerical order as indicated by the reviewer (page 12, line 8). The correction to the definition of E has been corrected in the revised manuscript as follows.

Page 12, lines 18–22: "... $V = 0.17 \text{ cm}^3$ is the breakdown-prone volume corresponding to $\hat{P} = P / P_{th}$, the ratio of the wave power to the threshold power in the focal plane ... $\frac{\Delta E}{\langle E \rangle}$ is the ratio of the energy required to produce one secondary electron–ion pair to that of the primary electrons generated by the gamma rays ...”

As indicated in the previous answer, the breakdown-prone volume was calculated to be 0.17 cm^3 . We thoroughly re-checked our calculations of the detectable masses of radioactive material 120 cm and 20 cm away from the source using the previously calculated breakdown-prone volume and confirmed them to be 2.4 mg and 0.067 mg, respectively. At 20 cm, the experimentally determined detectable mass indicates a sensitivity roughly 130 times greater than that predicted by the theoretical estimation (in the experiment, we used $0.5 \mu\text{g}$). At 120 cm, the proposed method becomes even more sensitive than predicted.

Comment 6: 6) Page 8 (lines 172-3): I recommend the authors to change their point of view on comparison of their method with the existing one. The method proposed by other authors is the first one and, as such, it proposes the simplest way to detect the presence of concealed radioactive materials (viz. the breakdown rate which depends on radioactive gammas). The authors propose a more delicate method based on measuring the difference in the delay time (essentially, the same as the authors of [2, 3]). Yes, this method has some advantages, but it also requires more accurate registration technique - and this is what should be explained.

Our response: We agree with the point raised by the reviewer. It might not be appropriate to apply the detectable mass equation directly in our method since our method allows for the statistical existence of free electrons in the breakdown-prone volume without the presence of

radioactive material. Therefore, we amended the sentences in the revised manuscript to include the fact that we compared our experimentally obtained detection sensitivities to the theoretically predicted values based on plasma on/off phenomena as follows.

Page 12, lines 8–10: “Now, we compare these experimentally obtained detection sensitivities to the theoretical results calculated based on plasma on/off phenomena.”

The key features of the proposed method are that the breakdown delay time decreases noticeably and that its statistical distribution disappears when radioactive material releases high-energy gamma rays. By measuring these phenomena (breakdown delay time reduction and disappearance of the statistical distribution of the delay time), the existence of radioactive material can be registered. Furthermore, during our additional experiments (performed during the revision of the paper), breakdown was also observed when the electric field was much weaker than the theoretical threshold electric field in ambient air in the presence of radioactive material. Without radioactive material, an incident RF power of more than 300 kW is required to initiate plasma breakdown; however, with radioactive material, an incident RF power of only 30 kW enabled plasma breakdown at 760 Torr in air. This experimental result indicates that the signal is strong, since plasma breakdown could be detected with an EM power much lower than the theoretical breakdown power at atmospheric pressure. Thus, the proposed method is a highly sensitive means of detecting radioactive material, as our experiments demonstrated.

A possible method of registering radioactive material can be described in detail as follows. First, according to the ambient air breakdown experiment with radioactive material, a single electron with an energy of around 770 keV is required to initiate plasma breakdown within the pulse length when the incident EM power is low (much lower than the threshold). (this analysis has been included in the revised manuscript along with the detailed calculations). Without radioactive material, it takes an infinite amount of time to achieve the required energy under low-power conditions. However, the radioactive material provides energetic electrons whose energies are greater than 770 keV and can be as high as 1 MeV, which enables the formation of breakdown within a given pulse length. Therefore, by determining the required incident EM power, it may be possible to estimate the energy of the electrons produced by the radioactive material. This information may be used to infer the type of radioactive material. However, it is not certain at this point that the initial energy of a single electron can be related to the type of radioactive material. Further experimental and theoretical studies including various quantities pertaining to radioactive materials and material types may reveal the relationship, which can then be employed to learn more about the registration technique.

Second, the ionization rate can be expressed in terms of the ionization enhancement factor α_{rad} and the ionization rate in ambient air (no radioactive material) Q_{rad} (typically 10–30):

$$\frac{dN_e}{dt} = \alpha_{rad} Q_{rad}$$

In the experiment (see page 7 of the revised manuscript), the plasma density was estimated to be around $6.44 \times 10^{13} / \text{cm}^3$, and the initial electron density (see page 19, Fig. 9 in the revised manuscript) was determined to be around 100 cm^{-3} based on the MCNP simulation. Therefore, a crude estimate of α_{rad} is around 2.7. From α_{rad} , it may be possible to estimate the quantity of radioactive material.

Comment 7: 7) Page 8 (line 175): I don't understand where the authors took the requirement to have the volume size less than 1 cubic mm: all what was required in previous studies is to have it less that 0.1 cubic cm, i.e. two orders more!

Our response: Thank you for the comment. We mistakenly reported the breakdown-prone volume to be 1 mm³ in the original manuscript. We have removed the wrong number from the revised manuscript.

Comment 8: 8) Page 9 (line 206). What is so surprising in the fact that the probability of the breakdown decreases as the distance from the radioactive source increases? In Ref. [10], Eq. (9) shows that the density of gammas ionizing air falls with the distance from the source faster than $1/R^2$. So, the question could be only: to what extent results shown in Fig. 6 of the MS agree with this dependence? This statement is also valid for the text on lines 234-5 about scalability of the results.

Our response: We appreciate the comment. In equation (9) in reference [17], if the distance from the radioactive source is much shorter than the range of propagation in air L_γ , the density of gamma quanta decreases approximately as $1/R^2$. In fact, the conditions of our experiment satisfy $R \ll L_\gamma$; therefore, the number of electrons decreases with $1/R^2$. The dependence of the number of electrons from our MCNP simulation agrees very well with the $1/R^2$ dependence, as demonstrated by Fig. 9b in the revised manuscript. We also observed that the probability decreases exponentially with the distance. We have added the fit curves for the average number of electrons and probability in Fig. 9, which is shown below.

Page 19, Fig. 9, lines 8–10: “c. Linear fit of the logarithm of the probability to d . The linear fits are in excellent agreement with the experimental data. The R-Square values represent how close the data are to the regression lines.”

Figure 9. Experimentally determined plasma breakdown probability and MCNP-simulation-based average number of free electrons as a functions of distance. Under normal conditions (i.e., in the absence of the radioactive source), plasma breakdown is not observable for an incident power of 28 kW in Ar at 400 Torr. **a.** Plasma formation probability, which decreases exponentially with increasing distance, and average number of free electrons, which decreases in proportion to $1/d^2$. The circles and diamonds represent the MCNP simulation results and the experimentally measured data, respectively, and the dotted curves are the fit curves. **b.** Linear fit of the average number of electrons to $1/d^2$. **c.** Linear fit of the logarithm of the probability to d . The linear fits are in excellent agreement with the experimental data. The R-Square values represent how close the data are to the regression lines.

 Reviewer #2 (Remarks to the Author):

Comment 1: The paper presents results of interesting experiments demonstrated how sensitive could be a remote detection of radioactive materials when using high power microwave radiation. These results could be interesting for readers. Unfortunately the paper text is written in too unclear and inaccurate manner. Some particular remarks are listed below. 1. The equation (3) is not justified at all. Moreover it seems to be wrong because it is quite different from the rigorous mathematical solution of the similar problem given in Ref.. 8.

Our response: Thank you for the comment. We have modified the manuscript by moving all of the equations to the Supplementary Information, in which the details of the equations are now provided, as the reviewer suggested. The derivation of equation (3) in the original paper has been inserted in the Supplementary Information, from the electron continuity equation to the equation for the probability of no breakdown (See Supplementary Note 3, equations (S17)–(S24)).

Comment 2: 2. A quantity $n(t)$ which is given in equation (2) is termed as averaged electron density but the quantity $n(t)$ is nondimensional. It should be noted that the quantity $n(t)$ is also presented as non-dimensional in Eq(1).

Our response: We appreciate the comment. The units of the average electron density should be particles per cubic centimeter (cm^{-3}). We have corrected the expression for the average electron density n (cm^{-3}) and added equation (S23) in Supplementary Note 3. With the initial electron density n_i being normalized to 1 in equation (S17), the average electron density \bar{n} (cm^{-3}) is given by equation (S23) in Supplementary Note 3, and the detailed derivation is as follows.

“We introduce the theoretical formative delay time derived from the electron continuity equation. The number of free electrons at time t is given by

$$n(t) = n_i \exp \left[\nu t \right] \quad (S17)$$

Here, n_i is the initial number of electrons, and $\nu = \nu_i - \nu_a - \nu_d$ is the net ionization rate in terms of ν_i , ν_a , and ν_d , which represent the ionization, attachment, and diffusion frequencies, respectively.

The probability of an avalanche reaching a size of N electrons based on equation (S17) is given by the expression^{4,5}

$$P(N) = \frac{1}{n} \left(\frac{N}{n} \right) \exp \left(-\frac{N}{n} \right) \quad (S22)$$

where \bar{n} (cm^{-3}) is the average value of N when $n_i = 1$ in equation (S17) and is given by

$$\bar{n} = \exp \left[\int_0^t \nu(t') dt' \right] \quad (S23)$$

Comment 3: 3. The quantity n_{cr} which is given in Eq(3) is also nondimensional and therefore it cannot be treated as critical plasma density which is measured in $1/\text{cm}^3$.

Our response: Thank you again for the comment. Our previous expression for the average electron density was not exactly well defined, as the reviewer suggested. The units should be cm^{-3} , as shown in equation (S23) in Supplementary Note 3. Therefore, the critical electron density should have units of cm^{-3} , like the plasma density. We have also corrected the expression for the probability of no breakdown. $P(n)dn$ in equation (3) in the original manuscript has been changed to $P(N)dN$ in equation (1) (page 9, line 4 in the revised manuscript and Supplementary Note 3, page 5, lines 4–9). The revised sentences in the manuscript are again as follows.

“...where \bar{n} (cm^{-3}) is the average value of N when $n_i = 1$ in equation (S17) and is given by

$$\bar{n} = \exp \left[\int_0^t (v_i(t') - v_d) dt \right] . \quad (\text{S23})$$

Upon combining equations (S22) and (S23), it becomes evident that, when the electron density approaches the critical density ($n_{cr} \approx 10^{14}/\text{cm}^3$), the plasma frequency is the same as the angular frequency ($f \approx 95$ GHz). Then, the probability of no breakdown can be expressed as

$$P(N < n, t) = \int_0^{n_{cr}} P(N) dN = 1 - \exp \left(-\frac{n}{n_{cr}} \right) . \quad (\text{S24})$$

Comment 4: 4. It is absolutely unclear how the theoretical line was obtained in Fig. 3.

Our response: Thank you for the comment. The detailed derivation of the probability of no breakdown has been inserted in Supplementary Note 3 (equations (S17)–(S24)). The final equation, equation (S24), shows the theoretical formative delay time distribution from $t = 0$ to the time at which the free electron density reaches the critical value. In the revised manuscript, the original Fig. 3 has been repositioned as Fig. 5 since we added to and replaced some of the original figures as part of the revision.

Comment 5: 5. A caption to Fig. 3 states that experimental results are obtained taking electric field amplitude close to the breakdown threshold. This is absolutely not sufficient because these results are very sensitive to a difference between the applied field amplitude and its breakdown value.

Our response: As the reviewer noted, the original Fig. 3 caption was somewhat vague. A clearer definition of the breakdown threshold electric field was needed. At each pressure, we

performed the experiment with the incident EM electric field being equal to the threshold field required for breakdown, which was determined to be the field at which 100% of the plasma would undergo breakdown. The experimental variance of the incident electric field at

the threshold was 0.15 kV/cm. We have modified the caption of Fig. 5 (the new figure number in the revised manuscript) to clarify the fact that the measurements were performed at the threshold electric field necessary for breakdown with a variance of 0.15 kV/cm. We have added more detailed descriptions of the experimental results shown in the new Fig. 5, as well as the following sentences.

Figure 5. Probability of no breakdown as a function of the delay time at pressures of **a.** 30 Torr, **b.** 100 Torr, and **c.** 250 Torr. The incident electric field at each pressure was the threshold field necessary for breakdown, which was defined as the field that would cause 100% of the plasma volume to undergo breakdown. The measured electric field variance was 0.15 kV/cm. The theoretical distributions were calculated using the Laue plots, as described in Supplementary Note 3.¹⁶ The red crosses and blue circles indicate the experimental data obtained with and without the radioactive source, respectively.

Page 6, lines 7–8: “Here, we defined the “threshold electric field” as the electric field at which 100% breakdown occurred in 200 shots experimentally.”

Page 7, Fig. 3, line 18: “The measured threshold electric field had 0.15 kV/cm variance.”

Page 9, lines 6–8: “The experiment used to obtain these data was performed with an electric field amplitude equal to the threshold value...”

Comment 6: 6. It is not clear what value the authors treats as the breakdown threshold which is shown in Fig. 5.

Our response: As mentioned in the previous response, the threshold electric field necessary for breakdown was defined as that at which 100% of the plasma volume would undergo breakdown. We have added more detailed descriptions in (page 6, lines 7–8 and page 7, Fig. 3, line 19) as follows.

Page 6, lines 7–8: “Here, we defined the “threshold electric field” as the electric field at which 100% breakdown occurred in 200 shots experimentally.”
 Page 7, Fig. 3, line 18: “The measured threshold electric field had 0.15 kV/cm variance.”

Figure 5 of the original manuscript can be elaborated more, as shown in the following figure.

Furthermore, during the revision of the manuscript, we performed extensive experimental work, including experiments at atmospheric pressure in Ar and air. Therefore, we have revised the theoretical Paschen curves and compared them with the experimental data. The original Fig. 5 is presented as Fig. 3 in the revised manuscript and is as follows.

Figure 3. Measured threshold fields for breakdown and theoretical Paschen curves for Ar and air. All of the points (blue crosses, red plus signs, and black circles) represent measured data, and the lines (black and blue) represent the theoretical curves. For Ar, measured data are depicted for the cases without radioactive material (blue crosses) and with 0.64 mCi of ^{60}Co (red plus signs). The measured threshold electric field had 0.15 kV/cm variance. The EM wave power was not sufficiently high to induce breakdown in the absence of the external radioactive source at pressures greater than 250 Torr in Ar. However, even though the output power of the gyrotron was insufficient to initiate avalanche ionization, saturation breakdown is observable even at 460 Torr,

due to the generation of free electrons by the ^{60}Co source. The maximum output electric field of the gyrotron, approximately 3.49 kV/cm, is below the required threshold electric field (~ 15.7 kV/cm at 760 Torr in Paschen curve) in Ar and air (black dotted vertical line). However, the observed plasma breakdown indicated by the black circles at 60 Torr and 760 Torr in air and 760 Torr in Ar present the possibility of plasma breakdown in the presence of radioactive material at powers lower than those predicted theoretically.

Comment 7: 7. The authors state several times that the radioactive source create some electron density which value decreases with an increase in the distance from the source. However this can be said so only when speaking about average electron density. The actual electron density fluctuates in time and these fluctuations plays the main role in the statistical fluctuations in the breakdown delay time which is studied in experiment.

Our response: We appreciate the reviewer's comment. The statistical delay time distribution is much wider when there is no radioactive material. In fact, the statistical nature of the occurrence of the free electrons is noticeably reduced when there is radioactive material nearby since the energetic "average" electron flux is maintained at a certain value at some distance. We have modified the sentence to clarify the meaning of the "average" electron density and added the following more detailed descriptions of the fluctuation of the actual electron density with time, as the reviewer suggested.

Page 3, Fig. 1, lines 10–12: "A high-intensity EM beam is irradiated near the hazardous source, resulting in instantaneous plasma breakdown at the focal point due to the high average number of free electrons generated by the radioactive material."

Page 18, lines 18–19: "The average number of free electrons in a volume of 1 cm^3 decreases with increasing distance, ..."

Page 18, lines 22–24: "The simulated average numbers of free electrons that are presented in Fig. 9b closely follow the expected trend."

Page 18, lines 25-27: "The probability of plasma breakdown becomes less than 20% at distances greater than 220 cm, at which the average number of free electrons is less than 2."

Page 19, Fig. 9, lines 2–3: "Experimentally determined plasma breakdown probability and MCNP-simulation-based average number of free electrons as a functions of distance."

Page 19, Fig. 9, lines 4–6: "a. Plasma formation probability, which decreases exponentially with increasing distance, and average number of free electrons, which decreases in proportion to $1/d^2$."

Page 19, Fig. 9, lines 7–8: "b. Linear fit of the average number of electrons to $1/d^2$."

Comment 8: 8. The authors predict very high sensitivity of their detection method in air based on the experimental results obtained in argon. But the breakdown process is air and argon are quite different and therefore the quantitative conclusion of authors seems to be incorrect.

The above list is not complete but to my mind it is quite sufficient to reject a publication of the paper in present form.

Our response: Thank you very much for your valuable comment. As noted, the original paper focused on the plasma breakdown process in Ar gas in the pressurized chamber. The

reason that we did not perform the experiment in ambient air was that the maximum output power of the laboratory-made gyrotron was insufficient. However, we fully agreed with the reviewer's opinion and tried to perform the experiment in ambient air since extrapolating the Ar experiment to ambient air is the most critical step. Since we considered this point to be very important, we revised the paper significantly after performing challenging additional experiments, including investigations of breakdown in atmospheric air in the presence of radioactive material. As mentioned, we initially thought that the experiment would be impossible in ambient air since the maximum allowable power of the millimeter-wave source (gyrotron) was 32 kW, which is 10 times lower than the power required to initiate plasma breakdown. However, very surprisingly, we were able to produce plasma breakdown even at this low power when the radioactive material was present, which is another observation that has not been reported previously. Based on the extensive plasma breakdown experiment in ambient air in the presence of radioactive material, we concluded that the statistical distribution of the delay time was also eliminated in air, as in Ar gas. Furthermore, we observed that the threshold electric field necessary for breakdown was significantly reduced in the presence of radioactive material. We significantly revised the whole manuscript by incorporating the plasma breakdown experiment in ambient air in the presence of radioactive material and added our interpretation of this observation, although additional quantitative theoretical analysis and experiments should be performed in the near future. Since the entire original manuscript has been revised significantly, we do not include here all of the modifications made regarding the experiments performed in ambient air. Thus, we refer the reviewer to the entire revised manuscript for our complete response to this comment.

Reviewer #3 (Remarks to the Author):

Comment 1: 1. This experimental study was motivated by the concept of remote detection of concealed radioactive materials by a focused terahertz (THz) radiation proposed by [V. L. Granatstein and G. S. Nusinovich, J. Appl. Phys. 108, 063304 (2010)].

2. According to this concept, a high-power THz radiation should be focused in a small spot where the field intensity exceeds the breakdown threshold. However, this volume should be so small that in the absence of ionizing sources in its vicinity the probability to have there any free electrons is low.

3. The current study was conducted by focusing of 95 GHz beam in Ar (which has a distinctly different chemistry than air) at almost diffraction limited, but even then only at pressures smaller than atmospheric.

4. The two plots that present the main results are intriguing to the experts, I wish they had been placed in a better-written paper.

Our response: Thank you for the comment. As suggested, we have modified the entire manuscript by adding experimental results regarding plasma breakdown in ambient air. The plots mentioned by the reviewer were reorganized in the revised paper (page 7, Fig. 3 and page 11, Fig. 6), and the explanations were elaborated upon as shown below.

Figure 3. Measured threshold fields for breakdown and theoretical Paschen curves for Ar and air. All of the points (blue crosses, red plus signs, and black circles) represent measured data, and the lines (black and blue) represent the theoretical curves. For Ar, measured data are depicted for the cases without radioactive material (blue crosses) and with 0.64 mCi of ^{60}Co (red plus signs). The measured threshold electric field had 0.15 kV/cm variance. The EM wave power was not sufficiently high to induce breakdown in the absence of the external radioactive source at pressures greater than 250 Torr in Ar. However, even though the output power of the gyrotron was insufficient to initiate avalanche ionization, saturation breakdown is observable even at 460 Torr, due to the generation of free electrons by the ^{60}Co source. The maximum output electric field of the gyrotron, approximately 3.49 kV/cm, is below the required threshold electric field (~ 15.7 kV/cm at 760 Torr in Paschen curve) in Ar and air (black dotted vertical line). However, the observed plasma breakdown indicated by the black circles at 60 Torr and 760 Torr in air and 760 Torr in Ar present the possibility of plasma breakdown in the presence of radioactive material at powers lower than those predicted theoretically.

Figure 6. Experimentally determined real-time ability to detect the presence of radioactive material at 19 kW and 250 Torr. **a.** Real-time measurement of the total delay time with and without the radioactive source. The source was in a lead box 20 cm from the focal point, and the lead box was opened and closed every 50 shots by an autocontrolled gate. The minimum delay times required for discharge with and without the source were approximately 2.2 μs and 4.1 μs, respectively. **b.** Delay time distributions measured with the ^{60}Co source 20–120 cm from the EM beam focal point. The delay time distributions obtained with and without the radioactive source are distinct from one another at each of the distances.

Comment 2: 5. No comparison with realistic air conditions (including impurities and weakly tight electrons) is supplied. Thus, the claim for the possible remote application is not supported by the current data and no discussion of how this data can be used in more realistic environment is presented

Our response: Thank you very much for your valuable comment. As noted, the original paper focused on the plasma breakdown process in Ar gas in the pressurized chamber. The reason that we did not perform the experiment in ambient air was that the maximum output power of the laboratory-made gyrotron was insufficient. However, we fully agreed with the reviewer's opinion and tried to perform the experiment in ambient air since extrapolating the

Ar experiment to ambient air is the most critical step. Since we considered this point to be very important, we revised the paper significantly after performing challenging additional experiments, including investigations of breakdown in atmospheric air in the presence of radioactive material. The experiments were performed in a realistic environment in which the temperature and humidity were maintained at 23°C ($\pm 0.5^\circ\text{C}$) and 62% ($\pm 2\%$), respectively. As mentioned, we initially thought that the experiment would be impossible in ambient air since the maximum allowable power of the millimeter-wave source (gyrotron) was 32 kW,

which is 10 times lower than the power required to initiate plasma breakdown. However, very surprisingly, we were able to produce plasma breakdown even at this low power when the radioactive material was present, which is another observation that has not been reported previously. Based on the extensive plasma breakdown experiment in ambient air in the presence of radioactive material, we concluded that the statistical distribution of the delay time was also eliminated in air, as in Ar gas. Furthermore, we observed that the threshold electric field necessary for breakdown was significantly reduced in the presence of radioactive material. We significantly revised the whole manuscript by incorporating the plasma breakdown experiment in ambient air in the presence of radioactive material and added our interpretation of this observation, although additional quantitative theoretical analysis and experiments should be performed in the near future. Since the entire original manuscript has been revised significantly, we do not include here all of the modifications made regarding the experiments performed in ambient air. Thus, we refer the reviewer to the entire revised manuscript for our complete response to this comment.

Comment 3: 6. Moreover, for remote applications the effect of the atmospheric turbulence on propagation and focusing of sub-THz wave beams in air should be considered. Even under very weak atmospheric turbulence, the effect of the turbulence is small only for distances up to 20m (1). At higher distances the required power will be well beyond the current technology.

7. In conclusion: This work can be of inherent interest to specialists from the remote detection of radioactive materials community and should be published in some forum. However, it not is a sufficiently strong case for publication in Nature Comm.

1. Nusinovich et al J. Appl. Phys. 113, 233303 (2013)

Our response: We appreciate the reviewer's comment. In a real situation, air turbulence may limit the remote detection range, as the reviewer noted. Air turbulence is difficult to predict and impossible to eliminate in ambient air. Temperature and humidity changes cause the absorption coefficient of the millimeter/THz wave to change as well (although the effect is weaker than it is at optical frequencies), and pressure changes may occur over the course of a day. However, local and instantaneous air turbulence may be eliminated by averaging multiple shots of the incident EM wave pulses by increasing the repetition rate of the incident wave pulse. Let us consider the effect of beam wander on the detection range. For the weak turbulence case, the distance R at which the turbulence is sufficiently weak to enable detection can be expressed as^A

$$[R(m)]^3 < \frac{10^{-2}[\rho_0(m)]^2}{0.434C_n^2 K},$$

where ρ_0 is the beam width; C_n^2 is the strength of turbulent refractive-index irregularities and is between $3 \times 10^{-13} \text{ m}^{-2/3}$ and $6 \times 10^{-17} \text{ m}^{-2/3}$, which correspond to strong and weak turbulence, respectively; and K is an integral coefficient ($\text{m}^{-1/3}$) defined in equation (32) in reference [A]. For our experimental setup, ρ_0 was around 5 mm. Under these conditions, R varies from 50 m to 1 km depending on the strength of the turbulent parameter. Therefore, it is advantageous for the wavelength to be longer since R depends on the size of the focused incident EM beam. From this estimate, we can conclude that the remote detection range can be extended to ~50 m and that in an environment with low turbulence (or clear and calm weather), the remote detection range may be increased up to 1 km with an antenna size of 1 m and a 95 GHz

source. As reported in reference [A], the beam wander time (on the order of milliseconds) is much longer than the incident EM wave duration (tens of microseconds); therefore, the wander of the beam during the pulse can be ignored. A discussion of the effect of air turbulence was added in the revised manuscript as suggested by the reviewer.

[A] Nusinovich, G. S., Qiao, F., Kashyn, D. G., Pu, R., & Dolin, L. S. Breakdown-prone volume in terahertz wave beams. *J. Appl. Phys.* **113**, 233303 (2013).

Reviewers' comments:

Reviewer #1 (Remarks to the Author):

I recommend to accept the revised MS for publishing in the journal after very minor revision in line with two comments:

1) in some places, the authors still do not distinguish the gyrotron power from the breakdown threshold defined by the RF field amplitude. For instance, the last sentence on page 4 ends as: "...the output power was lower than the minimum of required electric field amplitude". (See also the lines 280-281 on p. 13.)

2) I recommend checking of all reference numbers. For example, on p. 12, prior to Eq. (2), the authors mention ref. [17] and then, on the next line, Ref. [12]. It seems to me that in both cases the reference should be the same. Another example can be found on p. 19 around Eq. (5). Prior to Eq. (5), they mention Ref. [18], while after it they mention Ref. [17]. It is possible that in both cases they have in mind the same reference.

Reviewer #2 (Remarks to the Author):

The manuscript was certainly improved by the authors. However I am not quite satisfied even with new version of the paper. The paper text remains unclear and inaccurate. Below the main remarks are listed.

1. The authors still do not distinguish completely two qualitatively different quantities: (i) Electron number which is discrete and dimensionless and (ii) electron density which is continuous and is measured in $1/\text{cm}^3$. As the particular example I can refer a line which is presented just after Eq (S22) where the average electron number measured in $1/\text{cm}^3$ is related to initial electron number which equals 1.

2. The Poisson probability distribution (S22) is given without any comments. A reference to the Raizer book {Ref. 4} is not justified in this case because in this book the Poisson probability distribution was not applied to describe the breakdown avalanche. Moreover within the classical mathematical books the Poisson probability distribution is applied to discrete values whereas the authors use this distribution to describe the continuous quantity (electron density). Actually the Poisson distribution describe only a process of the seed electron production but the ionization, attachment and diffusion are also stochastic process which can modify the probability distribution as was shown in [Ref. 8].

3. After Eq (S24) authors discuss properties of such a quantity as the breakdown delay time.

"The theoretical breakdown formation time is a function of the pressure and the amplitude of the incident electric field. In the absence of an external radioactive source, the plasma avalanche occurs with a random delay time, which is referred to as the statistical delay time. The delay time decreases sharply in the presence of radioactivity due to the increase in the 75 average free electron density".

They also use this quantity in the Fig. 5 and within a caption to Fig. 5. However I have not found any definition (nor experimental not theoretical) of this quantity in the paper text.

4. Now in the paper text I have found the clear definition of the breakdown threshold. However the following sentence in a caption to the Fig. 5 is absolutely unclear.

"The incident electric field at each pressure was the threshold field necessary for breakdown, which was defined as the field that would cause 100% of the plasma volume to undergo

breakdown."

5. There is no detailed description concerning measurements of the breakdown probability shown on Fig. 5. 1.

To my mind the paper is not suitable for a publication in present form.

Reviewer #3 (Remarks to the Author):

The rewritten paper presents results of interesting experiments related to a remote detection of radioactive materials when using high power microwave radiation Unfortunately the paper text is still written in too unclear and inaccurate manner. In particular:

1. The abstract sounds as commercial advertisement and should be rewritten

2. Statement: "The detection sensitivity of this method was found to be much higher than that predicted by theoretical calculations based on plasma on/off phenomena; this increased sensitivity was achieved by precisely measuring the plasma breakdown delay time"

Comment: This sentence has to be modified, the precise measurement can't explain the theoretical discrepancy, the authors should point out the origin of this discrepancy.

3. Statement: "Health risks due to human-generated radioactivity, such as that released in nuclear power plant accidents and by nuclear weapons, have increased; in fact, such incidents are ultimately unavoidable"

Comment: For these applications the current detectors can be placed in the vicinity of the object.

4. Fig 3.

Comment: The deviation between the data obtained with and without radioactive material for the pressure below 250 mtor is marginal and without error bars can't justify the conclusions.

5. Statement: "As the pressure increases, the delay time increases accordingly due to the increase in the power required for plasma breakdown"

Comment: Please justify that the increase is correlated to the increase in power.

6. Statement "The considerable reduction observed in the measured threshold electric field (~ 16 kV/cm in theory...."

Comment: This sentence includes contradiction!

7. Statement: "Equation (3) indicates that when P/P_{th} is close to 1, the energy required to ionize one air molecule approaches infinity, assuming that a single electron gains its energy only from the incident EM field. For example, $\Delta\epsilon_{ion} = 1.6$ keV is necessary for $P/P_{th} = 1.5$. However, in the presence of radioactive material, an energy of up to ~ 1 MeV (in the case of ^{60}Co) is injected into the volume initially; therefore, it is probable that a significantly reduced RF power will induce plasma breakdown"

Comment: This paragraph includes values that contradict the well-established knowledge concerning ionization potentials Moreover, the authors can use the well-known information related to energy loss of 1MeV photons in the air.

8. Statement: "if the ionization energy of a single air molecule is 770 keV"

Comment: This statment includes value that contradicts the well-established knowledge concerning ionization potentials

9. Comment: The conclusions are written almost as commercial advertisement and must be rewritten

10. This paragraph includes values that contradict the well-established knowledge concerning ionization potentials

11. Statement: "Furthermore, we observed that the presence of radioactive material greatly reduced the breakdown threshold electric field in the atmosphere, although the reason for this observation is not completely understood at the present. Thus, further theoretical analyses should be performed in the future"

Comment: The analysis presented in the paper is highly incoherent and more modeling is required before considering publication in Nat. Com.

Reviewer #1

I recommend to accept the revised MS for publishing in the journal after very minor revision in line with two comments:

Comment 1: in some places, the authors still do not distinguish the gyrotron power from the breakdown threshold defined by the RF field amplitude. For instance, the last sentence on page 4 ends as: "...the output power was lower than the minimum of required electric field amplitude". (See also the lines 280-281 on p. 13.)

Our response: Thank you for your comment. In keeping with your recommendation, we have distinguished between the gyrotron output power and the breakdown threshold electric field in the revised manuscript.

Page 4, lines 22–24: "In addition, the air breakdown experiment was performed only with the radioactive source, even though the amplitude of the incident electric field was lower than the minimum required amplitude."

Page 14, lines 4-5: "However, the maximum possible electric field intensity for the gyrotron was lower than that according to the theoretical Paschen curve of air, as shown in Fig. 3."

Page 14, lines 8-9: "Of course, no plasma breakdown occurred in the absence of radioactive material in this electric field."

Comment 2: I recommend checking of all reference numbers. For example, on p. 12, prior to Eq. (2), the authors mention ref. [17] and then, on the next line, Ref. [12]. It seems to me that in both cases the reference should be the same. Another example can be found on p. 19 around Eq. (5). Prior to Eq. (5), they mention Ref. [18], while after it they mention Ref. [17]. It is possible that in both cases they have in mind the same reference.

Our response: Thank you for your comment. We have corrected the references in the revised manuscript. On page 13, Ref. [12] has been changed to Ref. [18], as the reference should be the same in both cases. (An additional reference was inserted before the previous Ref. [13] in the revised manuscript; therefore, it is now Ref. [18].) In addition, we have corrected the reference number of Ref. [17] to Ref. [19]. The following lines have been modified in the revised manuscript.

Page 12, line 14 to page 13, line 1: "The theory assumes that the incident EM beam is focused and that there is rarely a free electron present in the breakdown-prone volume in the

absence of radioactive material within the pulse duration of the incident EM wave, resulting in close to zero probability of breakdown. The theoretically detectable mass ($M(g)$) of ^{60}Co can be calculated using the following previously reported formula:¹⁸

Page 19, lines 21–23: “...where ρ_0 is the beam width; C_n^2 is the strength of turbulent refractive-index irregularities and is between $3 \times 10^{-13} \text{ m}^{-2/3}$ and $6 \times 10^{-17} \text{ m}^{-2/3}$, which correspond to strong and weak turbulence, respectively; and K is an integral coefficient ($\text{m}^{-1/3}$) defined in Ref. [19].”

Reviewer #2

The manuscript was certainly improved by the authors. However I am not quite satisfied even with new version of the paper. The paper text remains unclear and inaccurate. Below the main remarks are listed.

Comment 1: The authors still do not distinguish completely two qualitatively different quantities: (i) Electron number which is discrete and dimensionless and (ii) electron density which is continuous and is measured in $1/\text{cm}^3$. As the particular example I can refer a line which is presented just after Eq (S22) where the average electron number measured in $1/\text{cm}^3$ is related to initial electron number which equals 1.

Our response: We thank you for your comment. We have modified the relevant sentences describing the electron number density. Here are the changes made in the revised manuscript:

Page 9, lines 2-5: “where \bar{n} (cm^{-3}) is the average value of the number of electrons per volume. Further, the electron density approaches the critical density ($n_{cr} \approx 10^{14} \text{ cm}^{-3}$) at 95 GHz when the plasma frequency is the same as the angular frequency. $P_1(N)$ represents the formative delay time, which is the probability of an avalanche reaching a size corresponding to an electron density n .”

Supplementary document page 5, line 3: “The number density of free electrons at time t is given by...”

Supplementary document page 5, lines 5-7: “Here, n_i is the initial electron number density, and $\nu = \nu_i - \nu_a - \nu_d$ is the net ionization rate in terms of ν_i , ν_a , and ν_d , which represent the ionization, attachment, and diffusion frequencies, respectively.”

Supplementary document page 5, line 25: “...where \bar{n} (cm^{-3}) is the average value of N when $n_i = 1 \text{ cm}^{-3}$ in equation (S17) and is given by ...”

Comment 2: The Poisson probability distribution (S22) is given without any comments. A reference to the Raizer book [Ref. 4] is not justified in this case because in this book the Poisson probability distribution was not applied to describe the breakdown avalanche. Moreover within the classical mathematical books the Poisson probability distribution is applied to discrete values whereas the authors use this distribution to describe the continuous quantity (electron density). Actually the Poisson distribution describe only a process of the seed electron production but the ionization, attachment and diffusion are also stochastic process which can modify the probability distribution as was shown in [Ref. 8].

Our response: Thank you for your valuable comments. In our analysis of the breakdown probability, we only modeled the Poisson probability distribution for the breakdown avalanche while assuming that a seed electron exists initially in the breakdown-prone volume and that the number of electrons in the given volume increases exponentially, resulting in the formative delay time. In the analysis of the formative delay time, a stochastic process is not considered. When considering the existence of radioactive material near the breakdown-prone volume, it is assumed that high-energy primary electrons exist, which can be

considered the “seed electrons” in the volume. This assumption permitted the use of a Poisson distribution for describing the exponential increase in the number of electrons in the volume for estimating the formative delay time, even though the stochastic processes related to ionization, attachment, and diffusion were not considered.

Although the results of the formative delay time analysis presented in the original manuscript compared favorably with the experimental data, we found your comment invaluable in making the analysis more thorough. We agree that the probability can be determined with greater accuracy when the stochastic terms in Ref. [8] are included.

Therefore, we reanalyzed the experimental data for the probability distribution of the breakdown delay time by including a model of the stochastic delay time in the case of radioactive material. We followed the analysis procedure employed by Foster et al. and Dorozhkina et al. in Ref. [8,15] in the revised manuscript.

The statistical delay time in the presence of radioactive material can be obtained by fitting the experimental distribution to the Poisson distribution generated by the seeding electrons (S) as

$$P_2(n) = \frac{1}{n!} (S\Delta t)^n \exp(-S\Delta t),$$

(as also shown in equation (S25) in Supplementary Information). Here, S is the average rate of electron generation by the seeding source. In this study, it was the rate of electron generation owing to the primary electrons produced by the radioactive material. Further, as we had done already, the probability of no breakdown when considering only the formative delay time (in other words, the exponential avalanche time) is as follows:

$$P_1(N < n_{cr}, t) = \int_0^{n_{cr}} P(N) dN = 1 - \exp\left(-\frac{n_{cr}}{n}\right).$$

(as also shown in equation (S24) in Supplementary Information).

The total probability of no breakdown can be seen as the summation of the two probabilities: $P_1 + P_2$. We assumed that the source term, S , is independent of the pressure, as the primary electrons attributable to the radioactive material have high energies (the average energy is approximately 0.44 MeV). Therefore, the attachment rate of the source electrons to the molecules can be considered to be the same under all pressures. We fitted the formative and statistical distributions to the experimental data and compared them to the experimental results as follows. In the figure below, we have plotted the theoretical distribution while only taking the formative delay time into consideration (green solid line in figure below), as stated in the original manuscript. The black dotted curves represent the probability as a function of the delay time when the newly added statistical delay time is considered. As can be seen from the figure, taking the statistical delay time into consideration results in a better match with the experimental data.

Figure 5 in the original manuscript has been modified in the revised manuscript and now shows the curve for the theoretical probability. The modifications made to the revised manuscript and Supplementary Information are as follows:

Page 9, lines 6-28: “The probability distribution for the statistical waiting time for the avalanche such that zero electrons are found until per volume up to time, t , can be formulated

$$P_2(n=0, t) = \exp(-St), \quad (2)$$

where S is the average generation rate of initial electrons owing to the stochastic seeding source in the volume.¹⁵ We assume that the density of free electrons generated by the decay of ^{60}Co in the breakdown volume is constant and that the source term S is independent of the inner pressure of the chamber, as the primary seed electrons from the radioactive material have high average energies (approximately 0.44 MeV). Hence, the rate of attachment to the molecules can be considered to be the same under all pressures. By combining the two probabilities, that is, those for the formative and statistical delay times, the fitting curve can be obtained by changing the source S for the radioactive material case (black dashed line in Fig. 5) (see Supplementary Note 3).

Figure 5 depicts the probability of no breakdown as a function of the delay time. The experiment used to obtain these data was performed using an electric field with an amplitude equal to the threshold value necessary for plasma breakdown at each of the pressures. The black dashed lines indicate the theoretically calculated delay times described above. The delay times represented by the absence of the radioactive material (blue circles) increase as the pressure increases from 30 Torr to 250 Torr. It is evident that the statistical plasma formation delay time distributions extend to longer times in the absence of the radioactive

source. The measured minimum delay time in the presence of the radioactive material is less than 2.4 μs for each of the pressures, as shown in Fig. 5. At 250 Torr, not only the statistical delay time, but also the formative delay time, is reduced significantly in the presence of the radioactive source. These characteristics are strong indicators of the presence of a radioactive source nearby.”

Page 10, Fig. 5.:

Figure 5. Probability of no breakdown as a function of the delay time at pressures of **a.** 30 Torr, **b.** 100 Torr, and **c.** 250 Torr. At each pressure, the delay time was measured when the threshold RF electric field was applied. The threshold RF electric field amplitude was defined as the applied RF electric field amplitude at which 100% plasma breakdown occurred over 200 shots at a repetition rate of 1 Hz. The measured electric field variance was 0.15 kV cm^{-1} . The red crosses and blue circles indicate the experimental data obtained with and without the radioactive source, respectively. The theoretical curves (black dashed line) were calculated using the Laue plots in combination with the formative and statistical waiting times, as described in Supplementary Note 3.^{8,16,17}

Supplementary document page 6, lines 11-25: “The statistical delay time in the case where a radioactive material is present is calculated to fit the experimental results. This delay time indicates the period before the appearance of an initial electron to initiate the avalanche in the breakdown-prone volume. The Poisson distribution generated by the seeding source is given by^{3,7}

$$P_2(n) = \frac{1}{n!} (S\Delta t)^n \exp(-S\Delta t), \quad (\text{S25})$$

where S is the average rate of electron generation by the seeding source. We assume that the S term is independent of the inner pressure of the chamber owing to the creation of free electrons by the radioactive isotope. The probability of finding zero electrons ($n=0$) per volume up to time t is

$$P_2(n=0, t) = \exp(-St). \quad (\text{S26})$$

The source term, $S=6 \mu\text{s}^{-1}$, is empirically dependent on the background ionization rate owing to gamma-rays.

Therefore, the total delay time for plasma breakdown, described as the survival rate for a given pulse length t , is written as

$$P = P_1 + P_2. \quad (\text{S27})$$

Comment 3: After Eq (S24) authors discuss properties of such a quantity as the breakdown delay time.

“The theoretical breakdown formation time is a function of the pressure and the amplitude of the incident electric field. In the absence of an external radioactive source, the plasma avalanche occurs with a random delay time, which is referred to as the statistical delay time. The delay time decreases sharply in the presence of radioactivity due to the increase in the 75 average free electron density”.

They also use this quantity in the Fig. 5 and within a caption to Fig. 5. However I have not found any definition (nor experimental nor theoretical) of this quantity in the paper text.

Our response: Thank you for your comment. We have defined both the formative delay time and the statistical delay time in the revised manuscript and Supplementary Information as follows:

Page 4, lines 12-14: “The formative delay time is the time taken for the initial seed electron density to reach the critical plasma density at the operating frequency. The statistical delay time represents the period before the appearance of the initial seed electron.”

Supplementary document page 6, lines 6-7: “The theoretical breakdown formation time, defined in the manuscript, is a function of the pressure and amplitude of the incident electric field.^{3,6}”

Comment 4: Now in the paper text I have found the clear definition of the breakdown threshold. However the following sentence in a caption to the Fig. 5 is absolutely unclear.

“The incident electric field at each pressure was the threshold field necessary for breakdown, which was defined as the field that would cause 100% of the plasma volume to undergo breakdown.”

Our response: Thank you for mentioning this point. We have modified the caption for Fig. 5 so that it reads clearly.

Page 10, Fig. 5 caption: “At each pressure, the delay time was measured when the threshold RF electric field was applied. The threshold RF electric field amplitude was defined as the

applied RF electric field amplitude at which 100% plasma breakdown occurred over 200 shots at a repetition rate of 1 Hz. The measured electric field variance was 0.15 kV cm^{-1} .”

Comment 5: There is no detailed description concerning measurements of the breakdown probability shown on Fig. 5. 1.

To my mind the paper is not suitable for a publication in present form.

Our response: As per your recommendation, we have added a description of how the breakdown probability was measured, as follows:

Page 10, Fig. 5 caption: “At each pressure, the delay time was measured when the threshold RF electric field was applied. The threshold RF electric field amplitude was defined as the applied RF electric field amplitude at which 100% plasma breakdown occurred over 200 shots at a repetition rate of 1 Hz. The measured electric field variance was 0.15 kV cm^{-1} .”

Reviewer #3

The rewritten paper presents results of interesting experiments related to a remote detection of radioactive materials when using high power microwave radiation Unfortunately the paper text is still written in too unclear and inaccurate manner. In particular:

Comment 1: The abstract sounds as commercial advertisement and should be rewritten

Our response: Thank you for your comment. We have modified the abstract based on your comment. The revised manuscript is as follows.

Page 1, lines 13-26: “The remote detection of radioactive materials is impossible when the measurement location is so far from the radioactive source that the leakage of high-energy photons or electrons from the source cannot be sensed. In this study, we experimentally validated a detection method in which high-power pulsed electromagnetic waves at 95 GHz are used for the remote detection of radioactive material by inducing plasma breakdown. By measuring the plasma formation time and its dispersion, the detection sensitivity of the method could be enhanced significantly compared to the theoretically predicted sensitivity based merely on the plasma on/off phenomena. The elimination of the statistical nature of the delay time distribution in the presence of radioactive material was confirmed in pressurized Ar gas and air, and the dependence of the delay time on the power of the incident electromagnetic waves under different pressure conditions was evaluated. Furthermore, we experimentally observed that the incident electromagnetic wave power required for plasma breakdown in atmospheric-pressure air in the presence of radioactive material is significantly lower compared to that in the absence of radioactive material. An explanation for this experimental observation is proposed.”

Comment 2: Statement: "The detection sensitivity of this method was found to be much higher than that predicted by theoretical calculations based on plasma on/off phenomena; this increased sensitivity was achieved by precisely measuring the plasma breakdown delay time"
Comment: This sentence has to be modified, the precise measurement can't explain the theoretical discrepancy, the authors should point out the origin of this discrepancy.

Our response: Thank you for your comment. Because measuring the plasma breakdown delay time and its dispersion instead of the delay time results in improved detection sensitivity, the sentence that you refer to may have led to a misunderstanding. We have modified the statement in the abstract, so that it reads clearly now. The revised sentence is as follow:

Page 1, lines 17-22: “By measuring the plasma formation time and its dispersion, the detection sensitivity of the method could be enhanced significantly compared to the theoretically predicted sensitivity based merely on the plasma on/off phenomena. The elimination of the statistical delay time in the presence of radioactive material was confirmed in pressurized Ar gas and air, and the dependence of the delay time on the power of the incident electromagnetic waves under different pressure conditions was evaluated.”

Comment 3: Statement: "Health risks due to human-generated radioactivity, such as that released in nuclear power plant accidents and by nuclear weapons, have increased; in fact, such incidents are ultimately unavoidable"

Comment: For these applications the current detectors can be placed in the vicinity of the object.

Our response: Thank you for your comment. In the original manuscript, we had meant to suggest such accidents are sudden and unpredictable and that after such an accident, human beings cannot enter the region easily. We have modified the sentence so that our intended meaning is clear to the reader. The revisions made are shown below:

Page 2, lines 1-3: "Threats owing to man-made radioactivity, including accidents at nuclear power plants and nuclear weapons, have increased and are unavoidable. Once such an accident has occurred, the accident site should be closed to human beings."

Comment 4: Fig 3.

Comment: The deviation between the data obtained with and without radioactive material for the pressure below 250 mtor is marginal and without error bars can't justify the conclusions.

Our response: We thank you for your comment. As you mention, the discrepancy between the data obtained with and without radioactive material seems marginal because Fig. 3 shows the electric field for the 0–18 kV/cm range for all pressures, including the theoretical Paschen curve. In the figure below, we have magnified the Paschen curve to depict the data obtained with and without radioactive material. It is now possible to clearly distinguish between the data obtained with and without the radioactive material. Although we did not mention error bars in Fig. 3, they are referred to in the caption for Fig. 3:

Page 7, Fig. 3 caption: "The measured threshold electric field had 0.15 kV cm⁻¹ variance."

Comment 5: Statement: "As the pressure increases, the delay time increases accordingly due to the increase in the power required for plasma breakdown"

Comment: Please justify that the increase is correlated to the increase in power.

Our response: Thank you for your comment. The correlation between the delay time and the input power is shown in Fig 8 (e) in the original manuscript. As can be seen from the figure, as the pressure increases, the measured delay time increases from 0.67 μs to 1.12 μs at 32 kW. We also observed that the delay time decreased as the input power increased at 760 Torr.

Comment 6: Statement "The considerable reduction observed in the measured threshold electric field ($\sim 16 \text{ kV/cm}$ in theory...."

Comment: This sentence includes contradiction!

Our response: Thank you for your helpful comment. Indeed, this sentence is ambiguously worded. Therefore, we have corrected it as follows in the revised manuscript:

Page 17, lines 6-9: "The considerable reduction observed in the measured threshold electric field ($\sim 3.5 \text{ kV cm}^{-1}$ during the experiment performed in the presence of radioactive material versus $\sim 16 \text{ kV cm}^{-1}$ in theory in the absence of radioactive material) necessary to initiate plasma breakdown in air at atmospheric pressure in the presence of radioactive material requires further analysis."

Comment 7: Statement: "Equation (3) indicates that when P/P_{th} is close to 1, the energy required to ionize one air molecule approaches infinity, assuming that a single electron gains its energy only from the incident EM field. For example, $\Delta\epsilon_{ion} = 1.6 \text{ keV}$ is necessary for $P/P_{th} = 1.5$. However, in the presence of radioactive material, an energy of up to $\sim 1 \text{ MeV}$ (in the case of ^{60}Co) is injected into the volume initially; therefore, it is probable that a significantly reduced RF power will induce plasma breakdown"

Comment: This paragraph includes values that contradict the well-established knowledge concerning ionization potentials. Moreover, the authors can use the well-known information related to energy loss of 1 MeV photons in the air.

Our response: Thank you for your comment. We used Eq. (3) in the original manuscript to *qualitatively* explain the discrepancy between the experimental and theoretical RF power amplitude required for breakdown. In the original manuscript, $\Delta\epsilon_{ion}$ is defined as "the mean energy required for ionization of one air molecule" [1,2]. As you point out, $\Delta\epsilon_{ion} = 1.6 \text{ keV}$ for $P/P_{th} = 1.5$ is significantly higher than "the ionization potential energy," with those for nitrogen and argon being 15 eV and 12 eV, respectively. The reason a significantly greater amount of energy is required for the ionization of one air molecule is that most of the electron energy is spent on the excitation of the N_2 vibrational and electronic levels [2].

However, we have carefully reanalyzed the results. In order to explain the discrepancy between the experimental value and that expected based on theory more quantitatively, a field-reduction factor has been introduced; this is explained in detail in the answer to Comment 11 below. The manuscript has been modified based on the new analysis.

Comment 8: Statement: "if the ionization energy of a single air molecule is 770 keV"
Comment: This statement includes a value that contradicts the well-established knowledge concerning ionization potentials

Our response: Thank you for your comment. Again, we used Eq. (3) in the original manuscript to *qualitatively* explain the discrepancy between the experimental observation and theory in terms of the required RF power for breakdown. In the original manuscript, $\Delta\epsilon_{ion}$ is defined as "the mean energy required for ionization of one air molecule" [1,2]. As you point out, $\Delta\epsilon_{ion} = 1.6$ keV for $P/P_{th} = 1.5$, is significantly higher than "the ionization potential energy," with those for nitrogen and argon being 15 eV and 12 eV, respectively. The reason a greater amount of energy is required for the ionization of one air molecule is that most of the electron energy is spent on the excitation of the N_2 vibrational and electronic levels [2]. However, in order to explain the discrepancy between the experimental observation and theory, we have tried to analyze the results quantitatively by introducing the field-reduction factor, which is explained in detail in the answer to Comment 11 below. The manuscript has been modified based on the new analysis.

Comment 9: Comment: The conclusions are written almost as commercial advertisement and must be rewritten

Our response: Thank you for the comment. We have rewritten the conclusion in keeping with your comment. The modified conclusion is as follows:

Page 20-21, Conclusions: "The results of this experimental study confirm, for the first time, that it is possible to detect radioactive material remotely using a high-power millimeter-range gyrotron. We could observe the elimination of the statistical nature of the delay time distribution in the presence of radioactive material by using a 95 GHz, 30 kW gyrotron source. Owing to the plasma breakdown delay time measurements, the sensitivity of the proposed method in terms of the detectable mass was much higher (at least 130 times) than that predicted theoretically based on plasma on/off phenomena. The dependences of the delay time for plasma breakdown on the applied RF power and pressure were evaluated in Ar and air. The real-time remote detection of radioactive material was achieved in Ar and air at pressures of up to 760 Torr through delay time measurements. Furthermore, we observed that the presence of radioactive material greatly reduced the breakdown threshold electric field in the atmosphere; a quantitative analysis was performed to explain this observation by introducing an electric field-reduction factor. Compared to the currently available radioactive material detection technologies, the proposed technique allows for the detection of radioactive material at greater distances from the detector; detection at distances as large as 1 km should be feasible using an antenna of the proper size if the atmospheric turbulence is not strong."

Comment 10: This paragraph includes values that contradict the well-established knowledge concerning ionization potentials

Our response: Thank you again for the comment. As we have explained in the responses to Comments 7 and 8, we used Eq. (3) in the original manuscript to *qualitatively* explain the discrepancy between the experimental observation and theory in terms of the required RF power for breakdown. In the original manuscript, $\Delta\epsilon_{ion}$ is defined as “the mean energy required for ionization of one air molecule” [1,2]. As you point out, $\Delta\epsilon_{ion} = 1.6$ keV for $P/P_{th} = 1.5$ is much higher than “the ionization potential energy,” with those for nitrogen and argon being 15 eV and 12 eV, respectively. The reason the energy required for the ionization of one air molecule is much higher is that most of the electron energy is spent on the excitation of the N_2 vibrational and electronic levels [2]. However, in order to explain the discrepancy between the experimental observation and theory, we have tried to analyze quantitatively the results by introducing the field-reduction factor, which is explained in detail in the answer to Comment 11 below. The manuscript has been modified based on the new analysis.

Comment 11: Statement: "Furthermore, we observed that the presence of radioactive material greatly reduced the breakdown threshold electric field in the atmosphere, although the reason for this observation is not completely understood at the present. Thus, further theoretical analyses should be performed in the future"

Comment: The analysis presented in the paper is highly incoherent and more modeling is required before considering publication in Nat. Com.

Our response: Thank you for your valuable comment. We agree that a more quantitative analysis is necessary for interpreting the experimental data. Therefore, we have attempted to explain the experimental data based on the reduced electric field for breakdown in the presence of radioactive material more *quantitatively* [1].

Breakdown occurs when the electron density reaches the critical plasma density, which is

defined as $n_{cr} = \frac{\omega^2 m \epsilon_0}{e^2}$.

Therefore, the delay time for the occurrence of breakdown can be obtained under the condition

$$n_{cr} \leq n_0 e^{v_{eff,i} \tau} \quad (1)$$

$$\text{or, } v_{eff,i} \geq \frac{1}{\tau} \ln \left(\frac{n_{cr}}{n_0} \right). \quad (2)$$

The effective ionization rate, $v_{eff,i}$, depends on the amplitude of the RF field, E_0 . Therefore, one can express the functional formula for the ionization rate as

$$v_{eff,i}(E_0) = v_{am} Y \left(\frac{E_0}{E_{cr}} \right), \quad (3)$$

where E_{cr} is the critical field for inducing breakdown, and v_{am} is the typical dissociative attachment [1]. For $E_0 = E_{cr}$, $Y(1) = 1$, which means that the ionization rate is equal to the rate of attachment to the molecules [1]. If the amplitude of the applied RF field is significantly

greater than the critical field amplitude, then the ionization rate is higher than the attachment rate. This induces plasma breakdown.

From equation (3), one can get the inverse function of Y :

$$\frac{E_0}{E_{cr}} = Y^{-1} \left(\frac{v_{eff,i}(E_0)}{v_{am}} \right). \quad (4)$$

Because $Y = \frac{1}{\tau_p v_{am}} \ln \left(\frac{n_{cr}}{n_0} \right)$, the ratio of the threshold field to the critical field is

$$\frac{E_0}{E_{cr}} \propto \frac{1}{\tau_p v_{am}} \ln \left(\frac{n_{cr}}{n_0} \right). \quad (5)$$

Therefore, the threshold field (E_0) is inversely proportional to the pulse length (τ_p); this indicates that a longer RF pulse results in a decrease in the threshold field amplitude for plasma breakdown [3]. Further, E_0 depends on the logarithm of the ratio of the critical plasma density to the number density of the initial seed electrons.

Equation (5) provides insights into the decrease in the RF field required for breakdown (E_0) when a radioactive material is present. We postulate that the increased conductivity in the breakdown-prone volume leads to a decrease in the amplitude of the electric field for breakdown. We express the electric field required for breakdown in terms of the field-reduction factor, β , attributable to the presence of a radioactive material.

$$\beta E_0 = E_{cr}, \quad (6)$$

where $\frac{1}{\beta} = \ln \left(\frac{n_{cr}}{n_0} \right) / \ln \left(\frac{n_{cr}}{n_0^*} \right) = \ln \left(\frac{n_0^*}{n_0} \right)$. Here, n_0 is the seed electron number density when

there is no radioactive material and n_0^* is the seed electron number density in the presence of radioactive material.

The production rate of high-energy electrons and photons owing to 0.64 mCi ^{60}Co located 20 cm away as function of energy as calculated using MCNP code (MCNPX ver. 2.50).

The average number and energy of the high-energy electrons are approximately 50 and 0.44 MeV, respectively, as shown in Supplementary Fig. 2. Therefore, we can calculate the number of secondary knock-on electrons produced by a single high-energy electron as follows:

$$\frac{0.44 \text{ MeV}}{34 \text{ eV}} = 12600. \quad (7)$$

The time for the collision of a high-energy electron with a molecule can be estimated to be

$$t_{coll} = \frac{l}{v} = 1.6 \times 10^{-9} \text{ s}, \quad (8)$$

where l is the mean free path for the high energy electron, and v is the velocity of the electron. The mean free path for the high energy electron can be calculated as

$$l = \frac{1}{\sigma n}, \quad (9)$$

where σ is the scattering cross-section and n is the density of an air molecule. The scattering cross-section of an electron with an energy of 0.44 MeV is approximately 10^{-17} cm^2 [4], and n is approximately 10^{19} cm^{-3} at $T=300 \text{ K}$ and 1 atm pressure. Therefore, the mean free path for the high-energy electron is approximately $100 \text{ }\mu\text{m}$ and the collision time is approximately $4 \times 10^{-13} \text{ s}$. The total time for the generation of 12600 secondary knock-on electrons is approximately $5 \times 10^{-9} \text{ s}$.

Therefore, for a duration of $1 \text{ }\mu\text{s}$ before the plasma breakdown is induced owing to the applied RF pulse, the number density of the total secondary knock-on electrons generated by 50 high-energy electrons is approximately $1.3 \times 10^8 \text{ cm}^{-3}$.

A significant increase in the number of background free electrons owing to the high-energy electrons generated by the radioactive material results in an increase in the conductivity in the RF-focused volume, which, in turn, induces breakdown in the area. The dependency of the threshold electric field on the number of free electrons is shown in the figure below.

Electric field reduction factor, β , as function of background free electron density as calculated using equation (6).

The ratio of the electric field required for breakdown to the increased number of seed electrons under normal atmospheric conditions can be expressed as β . β gradually increases as the number of background free electrons increases. For an initial seed electron number density of 1.3×10^8 , β is approximately 2.5. In our experiments, we observed that the electric field required for breakdown decreased by a factor of 4 when the area was exposed to the radioactive material. Based on the analysis above, one can partly explain the reduction in the electric field required for breakdown (experimentally determined reduction factor of 4 versus theoretically determined reduction factor of 2.5), but the explanation is not complete. It should be noted that there have been experimental reports on atmospheric breakdown, with a similar mismatch in the experimental and theoretical results (theoretically determined reduction factor of 1.5). These mismatches had been attributed to uncontrollable experimental conditions such as the molecular contents and the presence of aerosol [1,5]. On the other hand, the reduction in the electric field required for breakdown has long remained a mystery with respect to “lightening” [6,7]. In the case of dry air, the measured electric field in lightning is lowered than the conventional electric field threshold for it [7]. It has been suggested that electrons with energy on the MeV scale result in a sustained avalanche that can lead to lightning at the electric field magnitudes much smaller than that needed for initiating breakdown [8]; this phenomenon is very similar to what was observed experimental in this study. Based on the analyses performed in [9,10], it is possible that a positive feedback loop attributable to the high-energy gamma rays aids the avalanche for breakdown, leading to a decrease in the electric field required. However, extensive simulation and experimental studies need to be performed to confirm this “positive feedback” theory.

The following modification have been made in the revised manuscript:

Page 17-18: “The considerable reduction observed in the measured threshold electric field ($\sim 3.5 \text{ kV cm}^{-1}$ during the experiment performed in the presence of radioactive material versus $\sim 16 \text{ kV cm}^{-1}$ in theory in the absence of radioactive material) necessary to initiate plasma breakdown in air at atmospheric pressure in the presence of radioactive material requires further analysis. We postulate that the increased conductivity in the breakdown-prone volume leads to the reduction in the electric field amplitude necessary for breakdown. We introduce a field-reduction factor, β , to express the reduced electric field required for breakdown owing to the presence of radioactive material:

$$\beta E_0 = E_{cr}, \quad (4)$$

where $\frac{1}{\beta} = \ln\left(\frac{n_{cr}}{n_0}\right) / \ln\left(\frac{n_{cr}}{n_0^*}\right) = \ln\left(\frac{n_0^*}{n_0}\right)$ (see Supplementary Note 4). Here, n_0 is the seed

electron density when there is no radioactive material; for typical atmospheric cases, a seed electron number density of $1\text{--}10 \text{ cm}^{-3}$ is widely accepted. n_0^* is the seed electron density in the presence of radioactive material, and n_{cr} is the critical plasma density at 95 GHz. Because the radioactive material generates high-energy gamma photons, high-energy electrons are produced in turn. The initial high-energy electrons with a density of n_{0e} produce secondary knock-on electrons via collisions with the molecules present in the breakdown-prone volume. To produce one secondary electron-ion pair, an average energy of $\sim 34 \text{ eV}$ is needed.¹⁴ The energy distribution of the gamma photons and high-energy electrons of 0.64 mCi ^{60}Co at 20 cm from the source can be determined using MCNP code (see Supplementary Figure 2). The calculated number of secondary knock-on electrons produced by a single high-energy electron is approximately 12600 (see Supplementary Note 4). Considering the pulse length of the applied RF field ($\sim 1 \mu\text{s}$) before the formation of the plasma, the total secondary knock-on electron number density attributable to the high-energy electrons is approximately 1.3×10^8

cm^{-3} (see Supplementary Note 4). The significant increase in the number of background free electrons owing to the high-energy electrons generated by the radioactive material results in an increase in the conductivity of the RF-focused volume, resulting in breakdown in the area. The field-reduction factor increases gradually as the number of background free electrons increases (see Supplementary Figure 3). For an initial seed electron number density of $1.3 \times 10^8 \text{ cm}^{-3}$, the calculated value of β is approximately 2.5, which means that the required electric field is reduced by a factor of 2.5 as compared to the field required when a radioactive material is not present. Based on this analysis, the reduction in the electric field required for breakdown (experimentally determined reduction factor of 4 versus theoretical reduction factor of 2.5) can be understood to a certain extent, although a slight discrepancy still exists between the experimental and theoretical observations. The additional reduction observed experimentally may be due to uncontrollable experimental conditions, including the fact that the exact molecular contents are unknown and an aerosol may be present.^{14,15}

In addition, Supplementary Note 4 has been to Supplementary Information.

References

1. Gurevich, A., Borisov, N., & Milikh, G., *Physics of Microwave Discharges: artificially ionized regions in the atmosphere* (CRC Press, 1997).
2. Nusinovich, G. S., Milikh, G. M. & Levush, B. Removal of halocarbons from air with high-power microwaves. *J. Appl. Phys.* **80**, 4189 (1996).
3. Gould, L. & Roberts, L. W. Breakdown of Air at Microwave Frequencies. *J. Appl. Phys.* **27**, 1162 (1956).
4. Phelps, A. V. & Pitchford, L. C. Anisotropic scattering of electrons by N_2 and its effect on electron transport. *Phys. Rev. A* **31**, 2932-2949 (1985).
5. Cook, A., Shapiro, M., & Temkin, R. Pressure dependence of plasma structure in microwave gas breakdown at 110 GHz. *Appl. Phys. Lett.* **97**, 011504 (2010).
6. Marshall, T. C., McCarthy, M. P. & Rust, W. D. Electric field magnitudes and lightning initiation in thunderstorms. *J. Geophys. Res. Atmos.* **100**, 7097-7103 (1995).
7. Dwyer, J. R., Smith, D. M. & Cummer, S. A. High-Energy Atmospheric Physics: Terrestrial Gamma-Ray Flashes and Related Phenomena. *Space Sci. Rev.* **173**, 133-196 (2012).
8. MacGorman, D.R. & Rust, W.D., *The electrical nature of storms* (Oxford University Press, 1998).
9. <http://physicsworld.com/cws/article/news/2013/apr/17/dark-lightning-sheds-light-on-gamma-ray-mystery>
10. Dwyer, J. The initiation of lightning by runaway air breakdown. *Geophys. Res. Lett.* **32** (2005).

In addition, a few additional modifications, apart from those made in response to the reviewers' comments, have been made in the revised manuscript. These are shown below:

(1) We have changed the probability scales from a logarithmic one to a linear one in Fig. 7 (a), Fig. 8 (a), and Fig. (b), so that they are similar to that in Fig. 5(a)–(c).

Page. 15, Figure 7 & Page. 16, Figure 8.

Figure 7. Experimental delay time measurements in the presence of radioactive material at different power levels in Ar gas at 760 Torr. **a.** Probability of no breakdown vs. delay time. **b.** Real-time measurements of delay time variations at output powers of 30 kW and 32 kW. No breakdown is observable in the absence of radioactive material, as indicated by the gray hatched regions.

Figure 8. Delay times experimentally measured in the presence of radioactive material at different EM output powers in air at 60 Torr and 760 Torr. **a, b.** Probability of no breakdown vs. delay time at 60 Torr and 760 Torr, respectively. **c, d.** Real-time delay time measurements obtained at 60 Torr and 760 Torr, respectively, along with 50% of the cumulative data measured with respect to the initial delay time. A gate on the lead box enclosing the radioactive source was opened and closed every 30 s by an autocontrolled gate. **e.** Dependence of delay time on pressure at various EM powers.

(2) A reference for the theoretical Paschen curve in air has been added in the revised manuscript:

Page 6, lines 4-5: “Then, we compared the measured values to the theoretical Paschen curves for Ar and air, as illustrated in Fig. 3.^{12,13}”

13. Löfgren, M., Anderson, D., Lisak, M. & Lundgren, L. Breakdown-induced distortion of high-power microwave pulses in air. *Phys. Fluids B-Plasma* **3**, 3528 (1991).

REVIEWERS' COMMENTS:

Reviewer #1 (Remarks to the Author):

I recommend accepting this MS. Indeed, there are some serious issues discussed by other referees which still deserve further discussions. Nevertheless, I think that publication of this MS even in its present form will cause an interest of the research community and stimulate theoretical and experimental studies of this important problem of remote detection of concealed radioactive materials by other research groups.

Reviewer#1 (commenting on Reviewer#2's concerns):

I went through the discussion of authors with the Reviewer #2 and came to the following conclusion.

Reviewer #2 made a valid point regarding the fact that the Poisson distribution can accurately describe the growth of electrons at the initial stage of the avalanche process only.

Then, the authors took into account the stochastic effects important in the processes associated with ionization, attachment etc and got simulation results which are much closer to the experimental data.

Therefore, to my mind, this issue is resolved: the authors are thankful to Referee and made some improvements of their paper.

I, however, would like to use this opportunity to make another recommendation, which is caused by their response to two last comments of Reviewer #2. Per his/her request the authors give the definition of the threshold RF electric field amplitude, in which they mention a number of shots at a repetition rate of 1 Hz required for 100% breakdown events.

I would also mention in this definition the shot duration, because for picosecond pulse duration the result can be different from that for the microsecond pulses.

Reviewer #3 (Remarks to the Author):

The manuscript was significantly improved

Reviewer #1

I recommend accepting this MS. Indeed, there are some serious issues discussed by other referees which still deserve further discussions. Nevertheless, I think that publication of this MS even in its present form will cause an interest of the research community and stimulate theoretical and experimental studies of this important problem of remote detection of concealed radioactive materials by other research groups:

Reviewer #2

Reviewer #2 made a valid point regarding the fact that the Poisson distribution can accurately describe the growth of electrons at the initial stage of the avalanche process only. Then, the authors took into account the stochastic effects important in the processes associated with ionization, attachment etc and got simulation results which are much closer to the experimental data. Therefore, to my mind, this issue is resolved: the authors are thankful to Referee and made some improvements of their paper. I, however, would like to use this opportunity to make another recommendation, which is caused by their response to two last comments of Reviewer #2. Per his/her request the authors give the definition of the threshold RF electric field amplitude, in which they mention a number of shots at a repetition rate of 1 Hz required for 100% breakdown events. I would also mention in this definition the shot duration, because for picosecond pulse duration the result can be different from that for the microsecond pulses.

Our response: We appreciate your valuable comment. We have added the 20 μ s pulse duration in the definition of the threshold RF electric field amplitude in the revised manuscript.

Page 4, lines 7-9: “Here, we defined the threshold electric field as the electric field at which 100% breakdown with 20 μ s pulse duration of 1 Hz repetition rate occurred in 200 shots experimentally.

Page 8, Fig. 3 caption: “The threshold RF electric field amplitude was defined as the applied RF electric field amplitude at which 100% plasma breakdown with 20 μ s pulse duration occurred over 200 shots at a repetition rate of 1 Hz.”

Reviewer #3

The manuscript was significantly improved.